# Multiomic characterization of pancreatic cancer-associated macrophage polarization reveals deregulated metabolic programs driven by the GM-CSF–PI3K pathway

**Seth Boyer[1†], Ho-Joon Lee[1*†‡], Nina Steele[2,3], Li Zhang[1], Peter Sajjakulnukit[1], Anthony Andren[1], Matthew H Ward[1], Rima Singh[4], Venkatesha Basrur[5], Yaqing Zhang[3], Alexey I Nesvizhskii[5,6], Marina Pasca di Magliano[3,7], Christopher J Halbrook[1*§], Costas A Lyssiotis[1,7,8*]**

[1]Department of Molecular & Integrative Physiology, University of Michigan, Ann Arbor, United States; [2]Department of Cell and Developmental Biology, University of Michigan, Ann Arbor, United States; [3]Department of Surgery, University of Michigan, Ann Arbor, United States; [4]Department of Molecular Biology and Biochemistry, University of California, Irvine, Irvine, United States; [5]Department of Pathology, University of Michigan, Ann Arbor, United States; [6]Department of Computational Medicine and Bioinformatics, University of Michigan, Ann Arbor, United States; [7]Rogel Cancer Center, University of Michigan, Ann Arbor, United States; [8]Department of Internal Medicine, Division of Gastroenterology and Hepatology, University of Michigan, Ann Arbor, United States

**\*For correspondence:**
ho-joon.lee@yale.edu (H-JL);
chalbroo@uci.edu (CJH);
clyssiot@med.umich.edu (CAL)

†These authors contributed
equally to this work

**Present address:** ‡Department
of Genetics, Yale Center for
Genome Analysis, Yale School of
Medicine, New Haven, United
States; §Department of Molecular
Biology and Biochemistry,
University of California, Irvine,
Irvine, United States

**Competing interest:** See page
19

**Reviewing Editor:** Elana J Fertig,
Sidney Kimmel Comprehensive
Cancer Center, Johns Hopkins
University, United States

**Abstract** The pancreatic ductal adenocarcinoma microenvironment is composed of a variety of cell types and marked by extensive fibrosis and inflammation. Tumor-associated macrophages (TAMs) are abundant, and they are important mediators of disease progression and invasion. TAMs are polarized in situ to a tumor promoting and immunosuppressive phenotype via cytokine signaling and metabolic crosstalk from malignant epithelial cells and other components of the tumor micro-environment. However, the specific distinguishing features and functions of TAMs remain poorly defined. Here, we generated tumor-educated macrophages (TEMs) in vitro and performed detailed, multiomic characterization (i.e., transcriptomics, proteomics, metabolomics). Our results reveal unique genetic and metabolic signatures of TEMs, the veracity of which were queried against our in-house single-cell RNA sequencing dataset of human pancreatic tumors. This analysis identified expression of novel, metabolic TEM markers in human pancreatic TAMs, including ARG1, ACLY, and TXNIP. We then utilized our TEM model system to study the role of mutant Kras signaling in cancer cells on TEM polarization. This revealed an important role for granulocyte–macrophage colony-stimulating factor (GM-CSF) and lactate on TEM polarization, molecules released from cancer cells in a mutant Kras-dependent manner. Lastly, we demonstrate that GM-CSF dysregulates TEM gene expression and metabolism through PI3K–AKT pathway signaling. Collectively, our results define new markers and programs to classify pancreatic TAMs, how these are engaged by cancer cells, and the precise signaling pathways mediating polarization.

## Editor's evaluation

This paper performs a comprehensive mechanistic and genomic evaluation of the impact of macrophage polarization on metabolic changes in pancreatic cancer. It provides an important advance to the understanding of the role of the microenvironment in the context of this disease.

## Introduction

Pancreatic cancer is the deadliest major cancer (*Siegel et al., 2020*). Early metastasis and insufficient detection methods compound an inability to effectively treat the disease, subjecting patients to a poor prognosis and high mortality rate (*Rhim et al., 2012*; *Chan et al., 2013*). The tumor microenvironment (TME), composed of a dense fibroinflammatory stroma, has been shown to contribute to the difficulty in treating this disease (*Feig et al., 2012*). In fact, the numbers of malignant cancer cells within pancreatic tumors are typically exceeded by the immune and fibroblast populations (*Feig et al., 2012*). Accordingly, recent efforts have sought to characterize these nonepithelial components of the TME in pursuit of identifying new and improved detection and treatment modalities. A predominant cell type of interest in the pancreatic TME are tumor-associated macrophages (TAMs), a myeloid cell population that mediates therapeutic resistance and disease aggression (*Di Caro et al., 2016*; *Halbrook et al., 2019*; *Zhang et al., 2017*; *Zhu et al., 2014*; *Zhu et al., 2017*; *Beatty et al., 2015*; *Candido et al., 2018*).

The impact of pancreatic TAMs on tumor growth and aggression has been relatively well established. As the major inflammatory component of solid tumors (*Balkwill and Mantovani, 2012*), TAM abundance correlates with worse response to pancreatic ductal adenocarcinoma (PDA) therapy (*Di Caro et al., 2016*). The mechanisms by which TAMs mediate this outcome are rather diverse. For example, TAMs promote cancer cell proliferation and metastasis (*Qian and Pollard, 2010*) and protect malignant cells from antitumor T-cell activity through immunosuppression (*Zhang et al., 2017*; *Candido et al., 2018*). TAMs have also been linked to promoting chemoresistance (*Zhu et al., 2014*), and recent work by our groups demonstrated that these pancreatic TAMs are capable of directly inhibiting the effect of chemotherapy agent gemcitabine on cancer cells through their release of the pyrimidine nucleoside deoxycytidine (*Halbrook et al., 2019*). These unique immunosuppressive and metabolic characteristics of TAMs are attributed in part to the phenotypic rewiring macrophages experience in response to the pancreatic TME.

TAMs have long been considered anti-inflammatory 'M2-like' macrophages, with in vitro models occasionally polarizing naive macrophages with type-2 cytokines to study TAMs (*Yuan et al., 2020*). Although overlap exists between the phenotypes of M2 and TAMs, including oxidative metabolism (*Halbrook et al., 2019*) and immunosuppressive properties, such as the expression of Arginase-1 (Arg1) (*Arlauckas et al., 2018*), the diverse molecular stimuli found throughout the TME polarize TAMs into macrophages with properties not shared with other classical subtypes. The focus of this study aimed to define the mechanistic aspects relating to TAM polarization by directly interrogating tumor cell–macrophage communication.

To recapitulate the signaling and metabolic factors present in the pancreatic TME, we polarized murine bone marrow-derived macrophages (BMDMs) in vitro with conditioned media from a PDA cell line in which we can regulate the activity of mutant Kras. We refer to macrophages polarized under these conditions as tumor-educated macrophages (TEMs) to distinguish them from TAMs arising in a tumor. We then utilized a systems biology approach integrating our multiomic profiling (i.e., transcriptomics, proteomics, metabolomics) to define biomarkers for, and the properties of, TEMs. Contrasting this with data from proinflammatory 'M1-like' and 'M2-like' macrophages revealed a panoply of markers and pathways that illustrate distinct functional characteristics of TEMs relative to classical subtypes. We then queried our in-house, single-cell RNA sequencing (scRNA-seq) datasets (*Steele et al., 2020*) and verified the expression of several of these markers in human pancreatic TAMs, demonstrating persistence of the TAM phenotype across different species and pancreatic cancer models.

Further inquiry into the role of cancer cell mutant Kras activity in TEM polarization led us to observe an important function of a Kras-driven signaling protein (i.e., granulocyte–macrophage colony-stimulating factor, GM-CSF) and a metabolite (i.e., lactate) for the expression of several unique TEM markers. Finally, we show that GM-CSF instructs TEM gene expression and metabolism through the PI3K–AKT pathway. Together, these data provide new insights into the crosstalk pathways between

cancer cells and macrophages and establish a mechanism by which malignant epithelial cells promote some of the most distinguishing features of TEM function.

## Results

### In vitro modeling and multiomic analysis of tumor-associated macrophages

To model pancreatic TAMs in vitro, we modified the classical BMDM differentiation and polarization paradigm (*Celada et al., 1984*), as follows (*Figure 1A*). First, we isolated and plated bone marrow in media containing macrophage colony-stimulating factor (M-CSF) to differentiate and expand macrophages for 5 days. These naive macrophages (M0) were then polarized to a tumor-associated phenotype for 2 days in conditioned media from PDA cells. Fresh media was included at a ratio of one to three parts conditioned media to account for nutrients consumed by the cancer cells. The resultant in vitro-derived cells are herein defined as TEMs, as they are educated by, and not directly associated with, cancer cells. Furthermore, the pancreatic cancer-conditioned media was generated from a cell line (iKras*3) derived from our murine pancreatic tumor model in which reversible mutant Kras expression is under the control of doxycycline (dox) (*Collins et al., 2012*). Growth of these cells in dox drives mutant Kras expression, MAPK pathway activity, and the malignant phenotype in vitro and in vivo (*Collins et al., 2012*; *Ying et al., 2012*). We also assessed how the removal of Kras from the cancer cells, via dox withdrawal for 3 or 5 days, impacted TEM polarization. In parallel with the TEM polarization strategies, we also polarized M0 macrophages into the canonical in vitro phenotypes with 2-day treatment of either lipopolysaccharide (LPS; proinflammatory 'M1') or interleukin-4 (IL4; anti-inflammatory 'M2'). M0 macrophages were maintained in the naive state by 2-day treatment with M-CSF (*Figure 1A*). M1 and M2 phenotypes were independently validated via quantitative polymerase chain reaction (qPCR) of classic proinflammatory (Interleukin 12b, *Il12b*; Tumor necrosis factor alpha, *Tnfa*) and anti-inflammatory (Found in inflammatory zone protein 1, *Fizz1*; Chitinase-like 3, *Chil3; Arg1*) genes (*Murray, 2017*; *Orecchioni et al., 2019*; *Figure 1—figure supplement 1A, B*). Importantly, we observed that TEMs did not fit into either the M1 or M2 marker profiles, suggesting that an unbiased approach would be needed to better define PDA-programmed macrophage populations.

We then performed multiomic profiling on each of the macrophage subtypes to achieve a comprehensive characterization of genetic and metabolic activity by (1) bulk RNA sequencing (RNA-seq) in triplicates, (2) proteomic profiling by mass spectrometry (MS) in duplicates, and metabolomic analyses on (3) intracellular and (4) extracellular metabolites by liquid chromatography (LC)/MS in triplicates. Principal component analysis (PCA) from each omics dataset demonstrated clustering of the biological replicates reflecting high-quality data (*Figure 1B*). From this global analysis, we also observed that the M1 subtype has the most distinct molecular profile on all triomics levels. The TEM subtype exhibited molecular profiles more similar to the M2 subtype than the M1, in line with previous publications from our groups and others (*Halbrook et al., 2019*; *Arlauckas et al., 2018*).

### Metabolism and cytokine signaling are distinctive features of pancreatic TEMs

As a means for further validation, we first directed our attention to known markers of each macrophage subtype in the transcriptomics data. We selected a group of five canonical macrophage genes, which were assessed in the primary data. The proinflammatory macrophage markers, Nitric Oxide Synthase 2 (*Nos2*) and the glucose transporter *Slc2a1* (GLUT1), displayed increased expression in LPS-treated macrophages compared to the other macrophage subtypes (*Figure 1C*, *Figure 1—figure supplement 1C*). Likewise, IL4-treated macrophages exhibited increased expression of classical anti-inflammatory/tissue remodeling markers, including Interleukin 4 Induced 1 (*Il4i1*), *Arg1*, and *Chil3* (*Figure 1D*, *Figure 1—figure supplement 1B*). Next, we performed differential expression or abundance analysis to identify markers that distinguish each subtype (*Figure 1E*, *Supplementary file 1*). The largest number of differential markers occurs in the M1 subtype across all triomics datasets, in agreement with the PCA analysis. We performed pathway analyses of each set of differential markers by gene set enrichment analysis (GSEA) (*Subramanian et al., 2005*) of KEGG gene sets for the transcriptomics data and Enrichr (*Xie et al., 2021*) for the proteomics data comparing TEMs to M0, M1, and M2 macrophages (*Figure 1—figure supplement 1D*, *Supplementary files 2 and 3*). Here, we

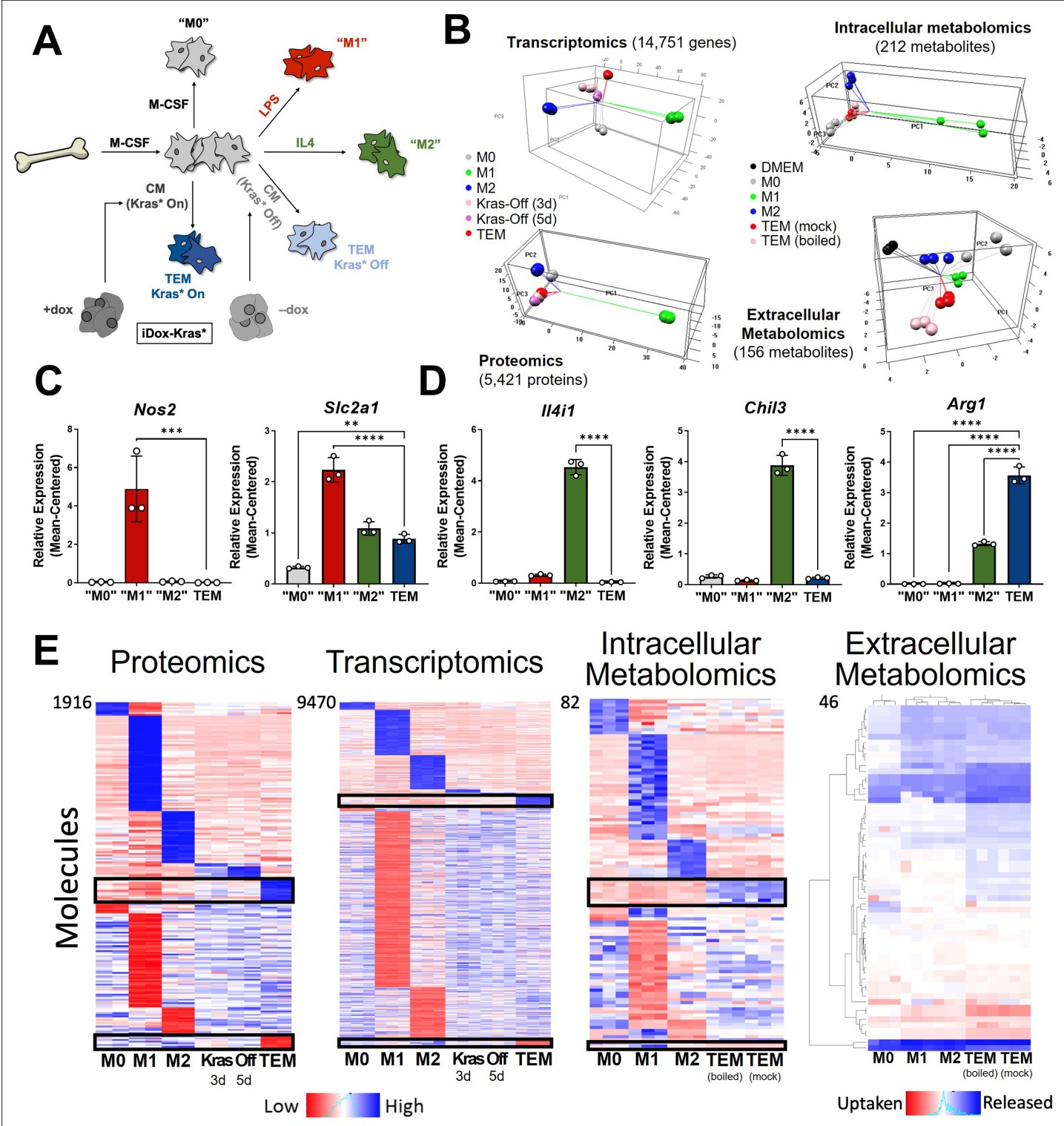

**Figure 1.** In vitro modeling and characterization of pancreatic tumor-educated macrophages (TEMs). (**A**) Schematic of bone marrow-derived macrophage (BMDM) differentiation and polarization. (**B**) Left: principal component analysis of transcriptomics and proteomics of BMDMs treated with macrophage colony-stimulating factor (M-CSF; M0), lipopolysaccharide (LPS; M1), interleukin-4 (IL4; M2), or conditioned media from Kras-Off (3 or 5 days) or Kras-On (TEM) pancreatic ductal adenocarcinoma (PDA) cells; right: intracellular and extracellular metabolomics from media (Dulbecco's modified Eagle medium, DMEM + 10% FBS), M0, M1, M2, or TEM (Kras-On media was mock treated or boiled before TEM culture). Transcriptomics and metabolomics samples were collected in biological triplicate; proteomics in biological duplicate. (**C**) RNA sequencing (RNA-seq)-measured mean-centered expression of classical M1 genes *Nos2* and *Slc2a1* across M0, M1, M2, and TEM phenotypes; $n = 3$. (**D**) RNA-seq-measured mean-centered

*Figure 1 continued on next page*

*Figure 1 continued*

expression of classical M2 genes *Il4i1*, *Chil3*, and *Arg1* across M0, M1, M2, and TEM phenotypes; *n* = 3. (**E**) Heat map array of differential markers of each subtype from proteomics (1916 proteins), transcriptomics (9470 transcript), and intracellular (82 metabolites) and extracellular (46 metabolites) metabolomics. We highlight TEM markers in the black boxes. Error bars in (**C**) and (**D**) are mean ± standard deviation (SD); significance comparisons are relative to TEM subtype and were calculated using one-way analysis of variance (ANOVA) with Dunnett's post hoc test; **p < 0.01, ***p < 0.001, ****p < 0.0001.

The online version of this article includes the following figure supplement(s) for figure 1:

**Figure supplement 1.** Transcriptomic analyses of macrophages markers.

observed several metabolic pathways that follow our previous characterization of TEM metabolism (*Halbrook et al., 2019*), including catabolic pathways (arginine and proline metabolism), anabolic pathways (nucleotide sugar metabolism), and functional pathways (fatty acid metabolism and glycolysis). This analysis also showed enrichment in the mTOR signaling pathway in TEMs, and the MAPK pathway in the other macrophage subtypes (*Figure 1—figure supplement 1D*). The top pathways among the upregulated TEM protein markers include neutrophil-related immune response and glycolipid/fatty acid metabolism.

Focusing further on the components driving TEM programming, pathway-centric approaches revealed two prominent features in TEMs relating to (1) cytokine signaling and (2) metabolism. Differential cytokine signaling is relatively well described for pancreatic TAMs (*Zhang et al., 2020*; *Aldinucci et al., 2020*). Indeed, C-C Motif Chemokine Receptor 1 (*Ccr1*) and *Ccr5* were significantly upregulated at the transcript level in TEMs, compared to M0, M1, and M2 macrophages (*Figure 2A*, *Figure 2—figure supplement 1A*). The patterns of differences in mRNA expression were maintained in the proteomics analysis (*Figure 2—figure supplement 1B*). Of note, our previous assessment of pancreatic TAMs identified CCR1 as a key mediator of immune suppression in pancreatic tumors (*Zhang et al., 2020*). Further, despite TAMs having long been described as M2-like/anti-inflammatory macrophages, due to their expression of ARG1 and oxidative metabolism (*Halbrook et al., 2019*; *Arlauckas et al., 2018*; *Binnemars-Postma et al., 2018*), pancreatic TEMs lack expression of important IL4 targets, demonstrating a clear difference in cellular activity between TEMs and type-2 cytokine-activated macrophages (*Figure 1D, E*, *Figure 1—figure supplement 1B*).

The second differentially enriched pathway in pancreatic TEMs is related to metabolism, and metabolic states have been shown to be key features distinguishing M1 and M2 macrophages (*Jha et al., 2015*). Indeed, by focusing on markers that are metabolic enzymes from both proteomics and transcriptomics, we find that TEMs contain the greatest proportion of upregulated metabolic enzymes, while M1 has the largest number of downregulated markers that are metabolic enzymes (*Figure 2B, C*).

Among the differential expressed TEM metabolic enzymes, we further narrowed our focus to three for follow-up analysis (*Figure 1C* and *Figure 2D, E*; *Figure 2—figure supplement 1C, D*). The first is Thioredoxin-interacting protein (TXNIP), an inhibitor of glucose import (*Lee et al., 2017*; *Wu et al., 2013*). *Txnip* emerged as the top upregulated TEM marker at both the gene and protein levels. The second was ATP Citrate Lyase (ACLY), a well-known enzyme with multifunctional roles in several biological pathways, including serving as a nexus between cellular metabolism and the regulation of gene expression by way of histone acetylation (*Zaidi et al., 2012*). Finally, we observed *Arg1* to be highly expressed in TEMs (*Figure 1D*), and even greater than that in 'M2-like' macrophages.

Next, we aimed to identify proteins that coexpressed with these three markers. We focused on correlated proteins, given the more proximal relevance to cellular functions and phenotypes than transcript expression. We selected the top 20 proteins according to both positive and negative correlations with ACLY, ARG1, or TXNIP (*Figure 2E*). Among those positively correlated with ACLY is SLC25A1, the mitochondrial citrate transporter. Citrate is a substrate of ACLY and highly abundant in TEMs based on our intracellular metabolomics data (*Figure 2—figure supplement 2A*), suggesting that the pathway of citrate-SLC25A1-ACLY is a TEM signature feature, as has been recently reported in inflammatory macrophages from atherosclerotic plaques (*Baardman et al., 2020*). Among the proteins positively correlated with ARG1 is *PIK3CD*, which endcodes for p110 delta, the catalytic subunit of PI3K (*Chen et al., 2019*; *Figure 2E*), suggesting a role for this signaling pathway in TEMs. We also investigated functional associations among those correlated proteins using Search Tool for Retrieval of Interacting Genes/Proteins (STRING) (*Szklarczyk et al., 2019*). A particularly strong

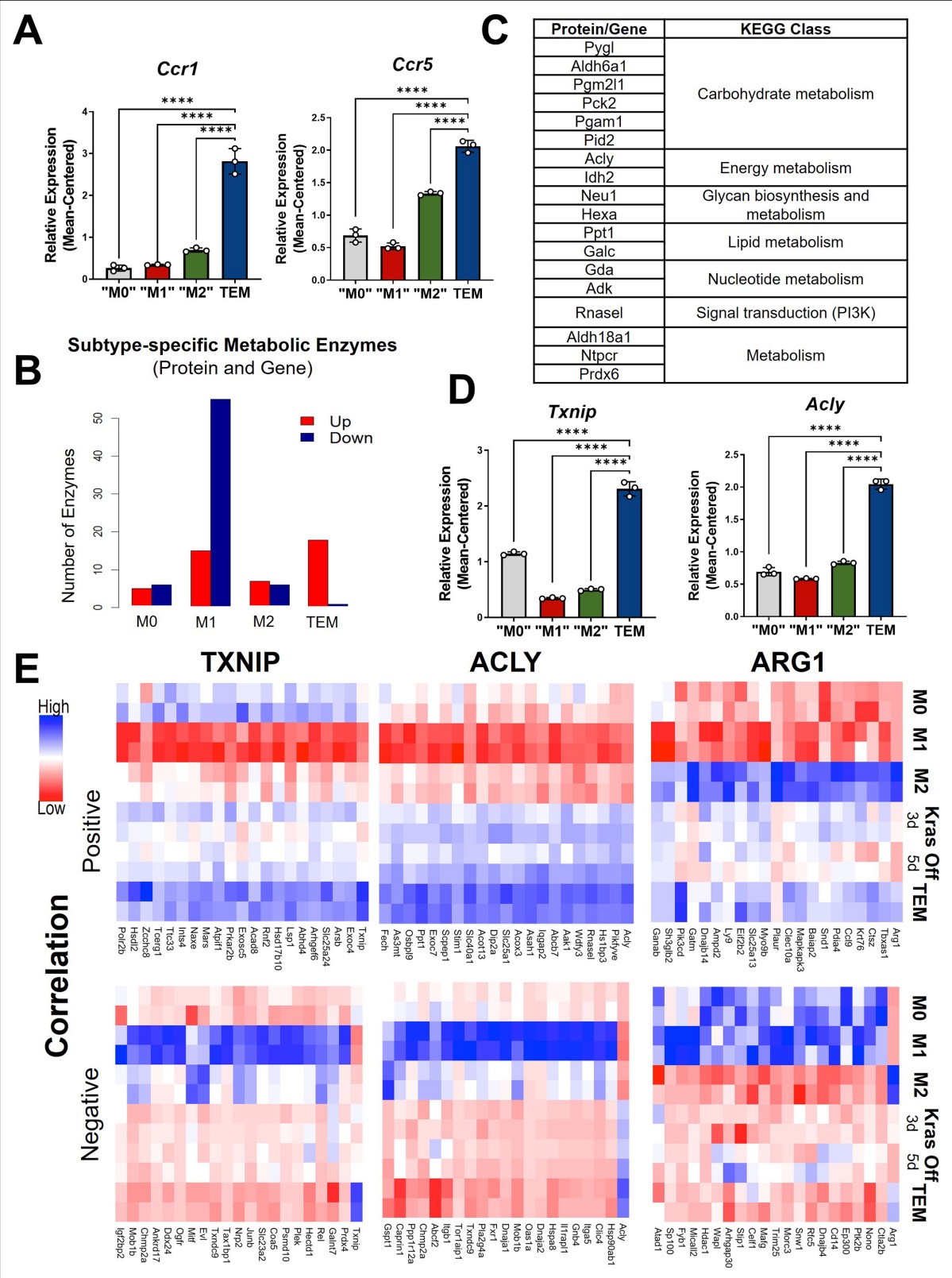

**C**

| Protein/Gene | KEGG Class |
|---|---|
| Pygl | Carbohydrate metabolism |
| Aldh6a1 | |
| Pgm2l1 | |
| Pck2 | |
| Pgam1 | |
| Pid2 | |
| Acly | Energy metabolism |
| Idh2 | |
| Neu1 | Glycan biosynthesis and metabolism |
| Hexa | |
| Ppt1 | Lipid metabolism |
| Galc | |
| Gda | Nucleotide metabolism |
| Adk | |
| Rnasel | Signal transduction (PI3K) |
| Aldh18a1 | Metabolism |
| Ntpcr | |
| Prdx6 | |

**Figure 2.** Metabolism and cytokine signaling are distinctive features of pancreatic tumor-educated macrophages (TEMs). (**A**) RNA sequencing (RNA-seq) mean-centered expression of TEM cytokine-related signatures, *Ccr1* and *Ccr5*; n = 3. (**B**) A bar plot of the numbers of up- and downregulated markers that are metabolic enzymes, present in both protein and gene analyses, for each subtype. (**C**) Table of 18 upregulated TEM markers from B and their corresponding KEGG (Kyoto Encyclopedia of Genes and Genomes) class. Note Acly as an enzyme of interest. (**D**) RNA-seq-measured mean-

*Figure 2 continued on next page*

Figure 2 continued

centered expression of TEM enzyme signatures *Txnip* and *Acly*; n = 3. (**E**) Heat map of the top 20 positively and negatively correlated proteins from the proteomics data for TXNIP, ACLY, and ARG1. Error bars in (**A**) and (**D**) are mean ± standard deviation (SD); significance comparisons are relative to TEM subtype and were calculated using one-way analysis of variance (ANOVA) with Dunnett's post hoc test; ****p < 0.0001.

The online version of this article includes the following figure supplement(s) for figure 2:

**Figure supplement 1.** Transcriptomic and proteomic analyses of macrophage markers.

**Figure supplement 2.** Multiomic pathway analyses.

functional association (enrichment p value ~0.0001) was found among the TXNIP-correlated proteins, which are mostly involved in metabolism (*Figure 2—figure supplement 2C*). This is not the case for those *Txnip*-correlated transcripts (*Figure 2—figure supplement 2D*).

## TEM markers distinguish pancreatic TAMs in human tumors

To demonstrate biological relevance of the pancreatic TEM phenotype, we queried our in-house scRNA-seq datasets from human tumors (*Steele et al., 2020*), paying particular attention to the myeloid populations (*Figure 3A*). We identified expression of several TEM markers in human pancreatic TAMs, such as *ACLY* and *TXNIP* (*Figure 3B*, *Figure 3—figure supplement 1A*). We are unable to provide sufficient data supporting *ARG1* expression in human pancreatic tumors as it experiences high rates of drop-out during scRNA-seq. However, we note expression of the strongly *Arg1*-correlated gene, *PIK3CD*, in macrophage populations in human pancreatic tumors (*Figure 3B*, *Figure 3—figure supplement 1A*).

In further support of PI3K relevance in TAMs, we found several PI3K-related TEM signatures (*Figure 2—figure supplement 2B*) also expressed in human TAMs (*Figure 3C*, *Figure 3—figure supplement 1B*). Those signature genes are indeed enriched in PI3K–Akt signaling pathway, as well as integrin signaling and the unfolded protein response by the Enrichr analysis (*Supplementary file 3*). These data suggest that PI3K signaling in TEMs is relevant in human TAMs, along with potential contributing factors both upstream and downstream of this signaling pathway.

## Pancreatic TAM polarization is dependent on mutant Kras activity in pancreatic cancer cells

The data from our profiling analyses revealed a distinction between the TEMs generated in media from Kras-expressing and -extinguished pancreatic cancer cells (*Figure 1B, E*). As noted in *Figure 1*, we polarized naive BMDMs with Kras-On and Kras-Off PDA cell-conditioned media (*Figure 4A*). Western blot of iKras*3 cell lysates for MAPK pathway activity demonstrated the expected decrease in ERK phosphorylation in dox-withheld iKras cells (*Figure 4B*). We turned our attention to differential markers in *Figure 1E* and their expression patterns in Kras-On and 5-day Kras-Off TEMs (*Figure 4C*). The data revealed broad differences in macrophage gene and protein expression, indicating that inducing mutant Kras in pancreatic cancer cells modifies both the extracellular environment and consequent phenotypes of macrophages exposed to these changes. Performing GSEA of KEGG gene sets between the Kras-On and Kras-Off macrophages (*Supplementary file 4*), we again observed enrichment in metabolic pathways, in line with our previous study of TEM metabolism (*Halbrook et al., 2019*). These include glycolysis, arginine catabolism, and pentose phosphate pathway. In addition to these metabolic pathways, we also see enrichment of the JAK–STAT pathway and activation of chemokine signaling/cytokine–cytokine receptor interactions. Furthermore, the top pathways among the protein markers in the Kras-Off condition include exosome/transport/apoptotic processes and macromolecule/nucleobase/phosphate metabolic processes (*Figure 4—figure supplement 1A* and *Supplementary file 3*). Specifically, we determined that macrophage expression of *Arg1*, *Acly*, *Txnip*, *Ccr1*, and *Ccr5* were all decreased when PDA cell Kras* was turned off (*Figure 4D*, *Figure 4—figure supplement 1B*).

## Mutant Kras activity in pancreatic cancer cells polarizes TEMs through GM-CSF and lactate

Upon recognizing the importance of PDA mutant Kras signaling for achieving the TEM phenotype, we considered the specific downstream factors of Kras activity that may contribute to TEM polarization.

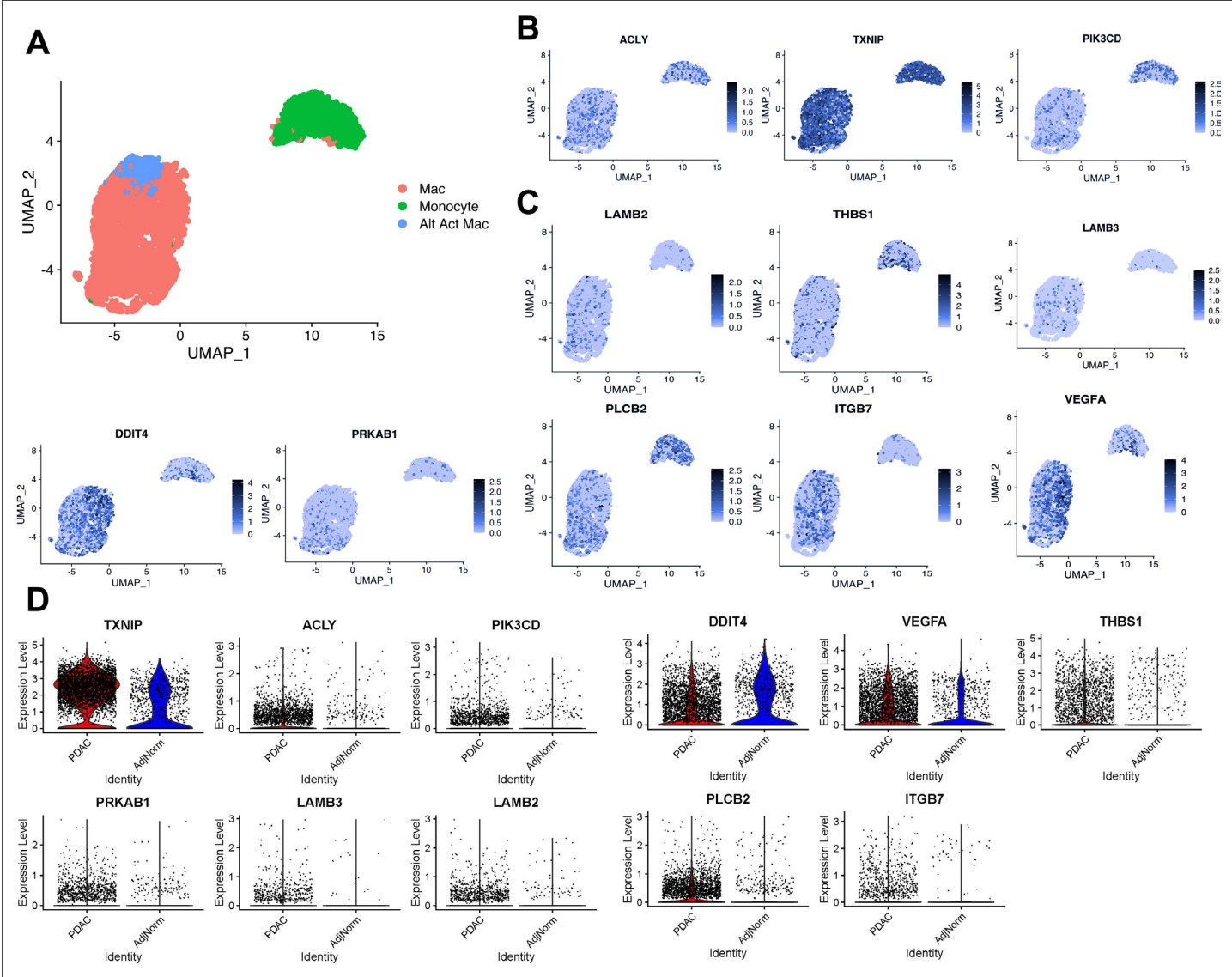

**Figure 3.** Expression of tumor-educated macrophage (TEM) markers in human pancreatic tumor-associated macrophages. (**A**) UMAP plot of myeloid populations in a human pancreatic tumor. (**B**) UMAP plots of TEM markers *ACLY*, *TXNIP*, and *PI3KCD* in human pancreatic tumor-associated macrophages (TAMs). (**C**) UMAP plots of PI3K-related genes expressed in human pancreatic TAMs (*DDIT4*, *PRKAB1*, *LAMB3*, *LAMB2*, *THBS1*, *VEGFA*, *PLCB2*, and *ITGB7*). (**D**) Expression of murine pancreatic TEM markers in macrophages from human tumors (pancreatic ductal adenocarcinoma [PDAC] TAMs) compared to macrophages from adjacent 'normal' tissue (AdjNorm), as analyzed by single-cell RNA sequencing.

The online version of this article includes the following figure supplement(s) for figure 3:

**Figure supplement 1.** Expression of pancreatic tumor-educated macrophage (TEM) markers in human tumors.

Macrophage expression of ARG1 and TXNIP has been shown to be responsive to lactate and extracellular acidification (*Zhang et al., 2019*; *El-Kenawi et al., 2019*). In addition, studies have demonstrated that macrophage expression of ARG1 may be regulated by signaling downstream of GM-CSF (*Jost et al., 2003*). Furthermore, previous work from our groups and others have also implicated mutant Kras activity in the activation of glycolysis and lactate excretion and the release of GM-CSF (*Ying et al., 2012*; *Tape et al., 2016*; *Bayne et al., 2012*; *Pylayeva-Gupta et al., 2012*).

Based on these leads, we determined if Kras expression promotes GM-CSF and lactate release in our isogenic, mutant Kras-inducible cell line model and the subsequent role of these factors on TEM polarization. Analysis of GM-CSF expression by qPCR and release by ELISA indicated that loss of mutant Kras expression reduced *Csf2* expression and GM-CSF release by more than 10- and 1000-fold, respectively (*Figure 5A*). Further, we found that GM-CSF secretion is abundant in two additional

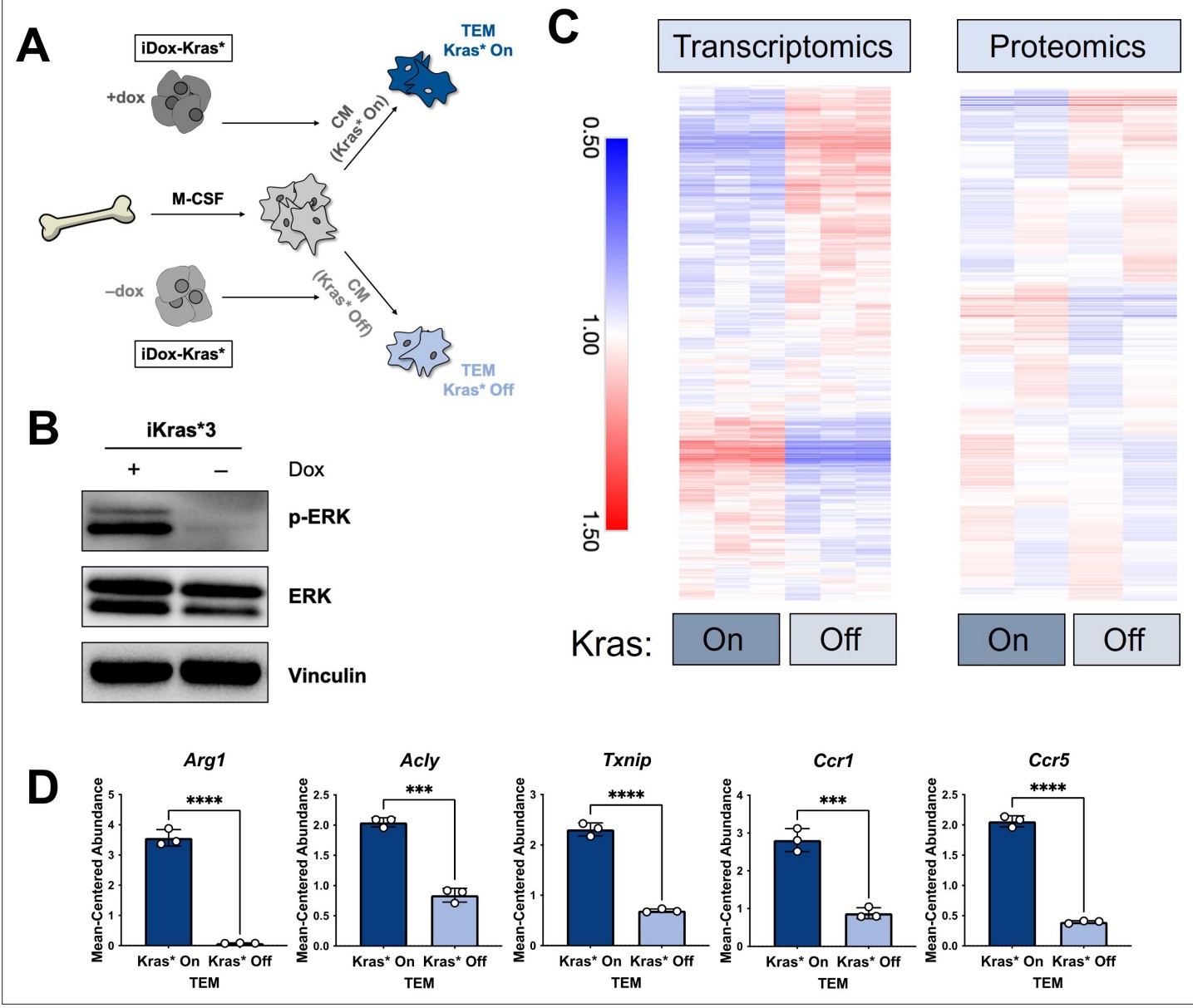

**Figure 4.** The polarization of pancreatic tumor-educated macrophages (TEMs) is dependent on mutant Kras signaling in pancreatic cancer cells. (**A**) Schematic of bone marrow-derived macrophage (BMDM) differentiation, iKras*3 cell Kras-On and Kras-Off conditioned media generation, and Kras-On and Kras-Off TEM polarization. (**B**) Western blot of MAPK pathway proteins ERK and pERK in Kras-expressing and 5-day Kras-extinguished iKras*3 cells. (**C**) Transcriptomics and proteomics heat maps of the differential markers in *Figure 1E* for Kras-On and 5-day Kras-Off TEMs. (**D**) RNA sequencing (RNA-seq)-measured mean-centered expression of TEM signatures Arg1, Acly, Txnip, Ccr1, and Ccr5; *n* = 3. Error bars are mean ± standard deviation (SD); significance was calculated using using a Student's *t*-test; ***p < 0.001, ****p < 0.0001.

The online version of this article includes the following figure supplement(s) for figure 4:

**Figure supplement 1.** Regulation of tumor-educated macrophage (TEM) gene expression by mutant Kras in pancreatic ductal adenocarcinoma (PDA) cells.

murine pancreatic cancer cell lines; that is, KPC cell lines, KPC7940 and KPCMT3 (*Figure 4—figure supplement 1C*). Next, we analyzed mutant Kras-dependent extracellular metabolism, including lactate production, by metabolomics. Metabolome profiling of the spent media from Kras-expressing PDA cells revealed profound alterations to the extracellular metabolome (*Figure 5—figure supplement 1A, B*), including a Kras expression-dependent increase in lactate release (*Figure 5B*). The

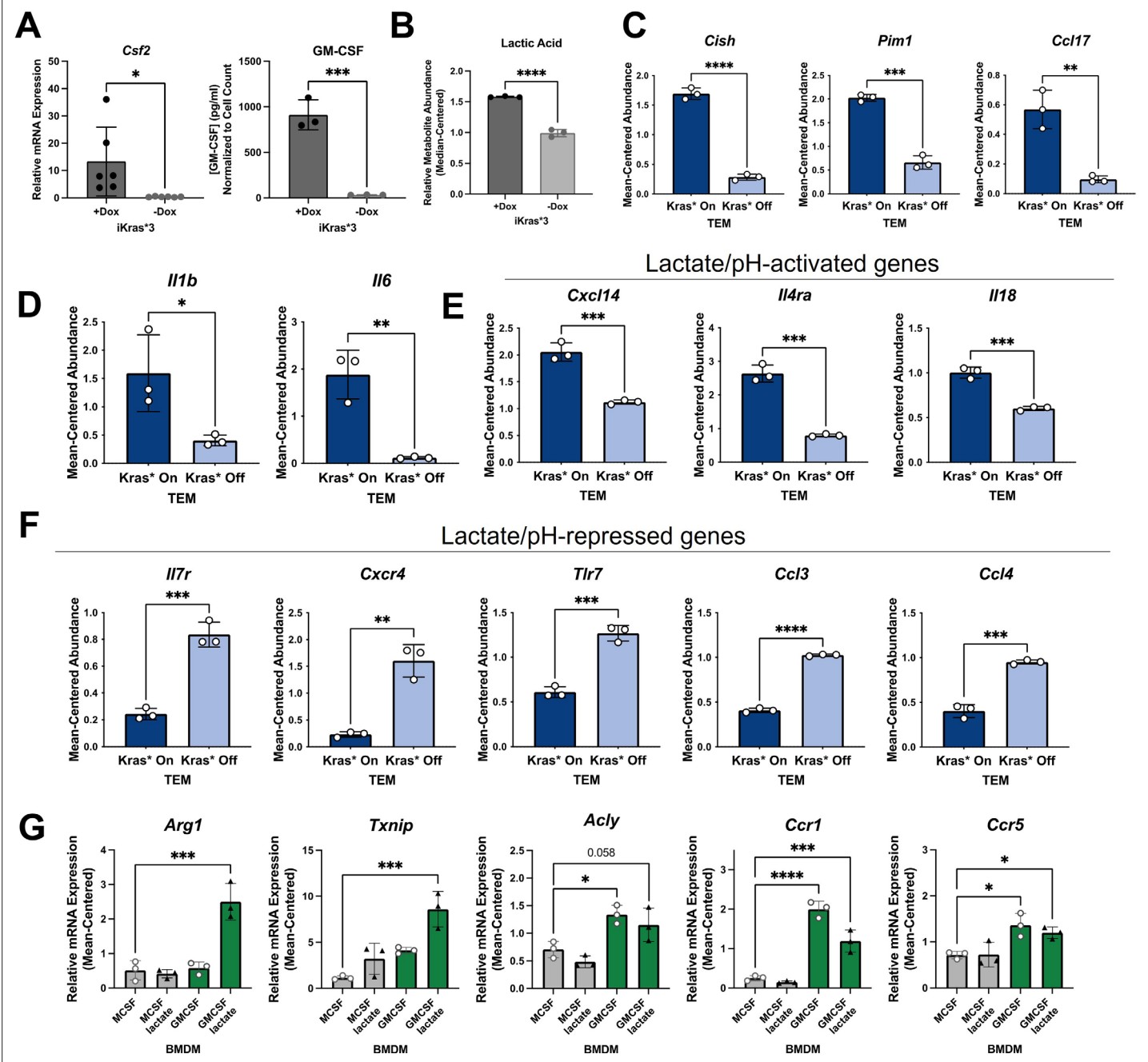

**Figure 5.** Kras in pancreatic ductal adenocarcinoma (PDA) polarizes pancreatic tumor-educated macrophages (TEMs) by way of granulocyte–macrophage colony-stimulating factor (GM-CSF) and lactate. (**A**) Quantitative polymerase chain reaction (qPCR) measurement of *Csf2* expression and ELISA for GM-CSF release; *n* = 3 in murine PDA cell line iKras*3. (**B**) Liquid chromatography (LC)/mass spectrometry (MS)-measured extracellular lactate abundance from Kras-expressing and -extinguished iKras*3 cells plotted as median-centered values; *n* = 3. (**C**) RNA sequencing (RNA-seq)-measured expression of myeloid GM-CSF-responsive genes, *Cish*, *Pim1*, and *Ccl17* in Kras-On and Kras-Off TEMs; *n* = 3. (**D**) RNA-seq-measured expression of lactate-responsive genes, *Il1b* and *Il6* in Kras-On and Kras-Off TEMs; *n* = 3. (**E, F**) RNA-seq-measured expression of genes responsive to acidic extracellular pH in Kras-On and Kras-Off TEMs; *n* = 3. (**G**) qPCR-measured expression of TEM markers *Arg1*, *Txnip*, *Acly*, *Ccr1*, and *Ccr5* in M0 macrophages treated with either lactate, GM-CSF, or the combination; *n* = 3. Error bars are mean ± standard deviation (SD); significance values in (**A–F**) were calculated using a Student's *t*-test; in (**G**), comparisons are relative to TEM subtype, and significance was calculated using one-way analysis of variance (ANOVA) with Dunnett's post hoc test; *p < 0.05; **p < 0.01, ***p < 0.001, ****p < 0.0001.

The online version of this article includes the following figure supplement(s) for figure 5:

**Figure supplement 1.** Effects of mutant Kras on extracellular metabolism in pancreatic ductal adenocarcinoma (PDA) cells.

Kras-dependent release of lactate was also analyzed and quantitated using an enzymatic assay-based approach (*Figure 5—figure supplement 1C*).

To confirm that these Kras-dependent differences in PDA cell activity impact activation of macrophages, we queried several differentially expressed genes in TEMs that are regulated by PDA mutant Kras and have been documented as responsive to GM-CSF, lactate, or pH. GM-CSF signaling in myeloid cells activates expression of *Cish*, *Pim1*, and *Ccl17* (*Lehtonen et al., 2002*). Indeed, we demonstrate that expression of these transcripts is significantly upregulated in macrophages exposed to Kras-On PDA cell-conditioned media, compared to those exposed to Kras-Off PDA cell-conditioned media (*Figure 5C*).

*Il1b* and *Il6* were previously identified as lactate-sensitive genes in macrophages (*Samuvel et al., 2009*). We demonstrate from our RNA-seq data that Kras-On media causes macrophages to express *Il1b* and *Il6* significantly more than macrophages exposed to Kras-Off media (*Figure 5D*). Furthermore, IL6 production has been shown to control macrophage Arg1 expression in an autocrine–paracrine manner (*Dichtl et al., 2021*), supporting the notion that Kras-dependent increases in lactate production impact the TEM phenotype.

Lactate is chiefly responsible for acidification of both the TME and the media used in tissue culture; the latter is well appreciated by the yellow shift of the pH-sensitive phenol red reagent. Multiple genes expressed by macrophages have been categorized as dependent on extracellular pH, with *Cxcl14*, *Il4ra*, and *Il18* shown to be increased in acidic extracellular conditions, and *Il7r*, *Cxcr4*, *Tlr7*, *Ccl3*, and *Ccl4* shown to be decreased in acidic conditions (*El-Kenawi et al., 2019*). Our RNA-seq data of Kras-On vs. Kras-Off TEMs support these patterns. Each of the aforementioned genes that are increased in acidic conditions are upregulated in Kras-On TEMs, and each of these genes that are decreased in acidic conditions are downregulated in Kras-On TEMs (*Figure 5E, F*). These data collectively support the hypothesis that increased cancer cell production of GM-CSF and lactate is dependent on mutant Kras, and that the differential production of mutant Kras-dependent factors modifies both the extracellular environment and subsequent phenotypes of neighboring macrophages.

To further confirm that extracellular GM-CSF and lactate are important contributors to the TEM phenotype, we treated naive macrophages (M0) for 48 hr with either GM-CSF, lactate, or the combination. A naive macrophage culture was maintained with M-CSF as a control. In order to more closely mimic the cancer cell-conditioned media, and to assess effects that lactate-induced extracellular acidity may have on TEM polarization, we maintained media supplemented with lactate at a lower pH. We then collected cell lysates and performed qPCR for our TEM markers. These results demonstrated that *Arg1* and *Txnip* are not significantly increased by GM-CSF or lactate as independent treatments. In contrast, these two factors in combination impose a synergistic effect on the expression of both genes (*Figure 5G*). We also identified increases in *Acly*, *Ccr1*, and *Ccr5* expression in macrophages treated with GM-CSF. These results, in combination with the increased levels of lactate and GM-CSF observed in Kras-On media, provide strong supporting evidence for the essential role of these factors in TEM polarization.

## PDA-derived GM-CSF promotes TEM polarization through the PI3K–AKT pathway

GM-CSF has pleiotropic effects on signal transduction, dependent on signal strength and context (*Zhan et al., 2019*; *Hamilton, 2019*). Classic downstream pathways activated by GM-CSF include NF-κB, PI3K/AKT, and the MAPK pathway. The data presented in *Figure 2E* and *Figure 2—figure supplement 2B* suggested that TEM polarization was marked by an increase in the PI3K pathway. To test the role of PI3K signaling in TEM polarization downstream of GM-CSF, we activated BMDMs with Kras-On media in the presence or absence of either the pan-PI3K inhibitor, BKM120, the pan-AKT inhibitor, MK-2206, or vehicle control. Western blotting for ARG1 revealed a strong activating role for the PI3K/AKT pathway in pancreatic TEMs (*Figure 6A*). AKT is known to phosphorylate ACLY, which has been shown to then modify histone acetylation that impacts Arg1 expression in IL4-stimulated macrophages (*Covarrubias et al., 2016*). In support of these findings, we also demonstrate that PI3K/AKT inhibition decreases ACLY phosphorylation in our TEM model (*Figure 6A*, *Figure 6—figure supplement 1A*).

Because GM-CSF is secreted by Kras-expressing PDA cells, impacts macrophage Arg1 expression (*Park et al., 2019*), and activates PI3K (*Hamilton, 2019*), we postulated that cancer cell GM-CSF may

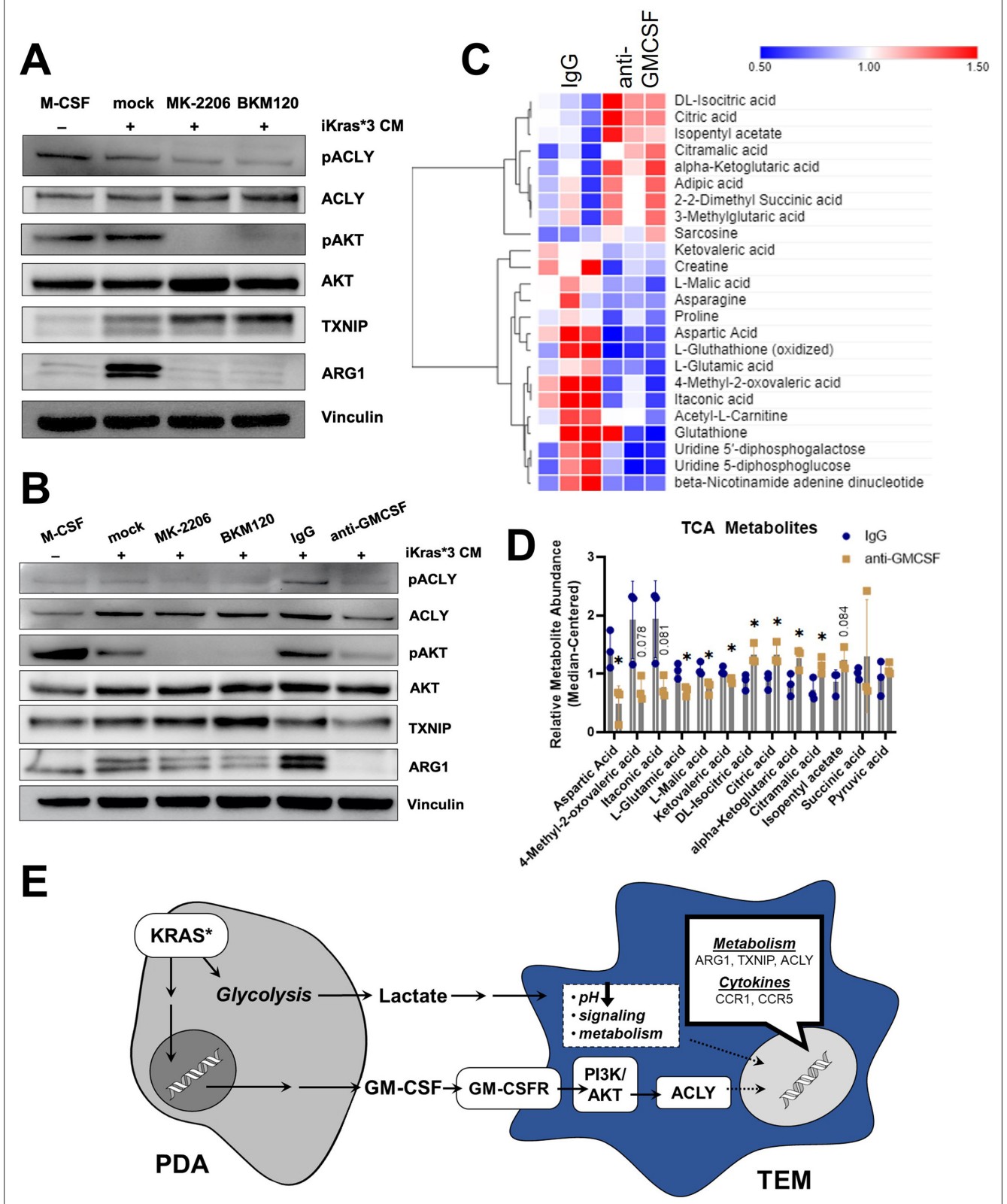

**Figure 6.** Pancreatic ductal adenocarcinoma (PDA)-derived granulocyte–macrophage colony-stimulating factor (GM-CSF) promotes tumor-educated macrophage (TEM) polarization and dictates their metabolism through the PI3K–AKT pathway. (**A**) Western blot of pACLY/ACLY, pAKT/AKT, TXNIP, and ARG1 in bone marrow-derived macrophages (BMDMs) treated with either macrophage colony-stimulating factor (M-CSF), iKras*3 cell-conditioned media + vehicle, iKras*3 cell-conditioned media + pan AKT inhibitor MK-2206, or iKras*3 cell-conditioned media + pan-PI3K inhibitor BKM120. (**B**)

*Figure 6 continued on next page*

*Figure 6 continued*

Western blot of pACLY/ACLY, pAKT/AKT, TXNIP, and ARG1 in BMDMs treated with either M-CSF, iKras*3 cell-conditioned media + vehicle, iKras*3 cell-conditioned media + MK-2206, iKras*3 cell-conditioned media + BKM120, iKras*3 cell-conditioned media + IgG control, or iKras*3 cell-conditioned media + GM-CSF-neutralizing antibody. (**C**) Heat map of differentially abundant metabolites in TEMs treated with either anti-GM-CSF or IgG control; $n = 3$. (**D**) Bar graph of TCA and related metabolites in TEMs treated with either anti-GM-CSF or IgG control; $n = 3$. (**E**) Schematic of TEM polarization model. Replicates and quantitation of the westerns in (**A, B**) are presented in *Figure 6—figure supplement 1A*. Error bars are mean ± standard deviation (SD); significance was calculated using a Student's *t*-test; *$p < 0.05$.

The online version of this article includes the following figure supplement(s) for figure 6:

**Figure supplement 1.** Granulocyte–macrophage colony-stimulating factor (GM-CSF) neutralization disrupts the tumor-educated macrophage (TEM) metabolic phenotype.

be activating macrophage Arg1 expression through the PI3K–AKT pathway. To test this hypothesis, we treated BMDMs with Kras-On conditioned media and either a GM-CSF-neutralizing antibody or IgG control. Indeed, blocking GM-CSF resulted in a dramatic decrease in ARG1 expression, as measured by immunoblotting (*Figure 6B*). GM-CSF neutralization also resulted in decreased phosphorylation of both AKT and ACLY, confirming that GM-CSF activates macrophage Arg1 expression through PI3K signaling (*Figure 6B*, *Figure 6—figure supplement 1A*). Finally, GM-CSF neutralization also led to a modest decrease in expression of ACLY and TXNIP (*Figure 6B*, *Figure 6—figure supplement 1B*). Taken together, these data support an activating role of GM-CSF on TEM polarization.

Lastly, to analyze how these changes in gene expression impact metabolism, we used our LC/MS-based metabolomics profiling approach in TEMs treated with anti-GM-CSF relative to control antibody (*Figure 6C*). The anti-GM-CSF-treated groups displayed an increase in citrate, potentially reflecting decreased ACLY activity, in treated cells (*Figure 6D*). In contrast to citrate, other TCA cycle and associated metabolites, including malate, itaconate, glutamate, and aspartate, were decreased following GM-CSF neutralization (*Figure 6D*), which suggests that GM-CSF blockade disrupts the TCA cycle and metabolism of associated amino acids. Collectively, these data demonstrate that TEMs are functionally coordinated by GM-CSF stimulation of PI3K signaling in order to maintain their metabolic homeostasis (*Figure 6E*).

## Discussion

The pancreatic TME consists of a heterogenous mixture of cells and extracellular matrix. TAMs are one of the most abundant cell types in PDA and participate in therapeutic resistance through a variety of mechanisms, including resistance to chemotherapy, immunosuppression, and promotion of tumor growth. However, the factors that contribute to the unique functional properties of TAMs remain insufficiently characterized.

Here, we employed a multiomic approach to molecularly define pancreatic TAMs. Bulk RNA sequencing, MS-based proteomics, and LC/MS-based metabolomics revealed several distinctions between TEMs and classical macrophage subtypes. Our focus on metabolism and cytokine signaling as two primary drivers of cellular function revealed Txnip, Acly, and Arg1 as unique contributors to TEM metabolism. The top 20 proteins correlated with Txnip showed enrichment in metabolism. Acly revealed strong correlation with Slc25a1, the mitochondrial citrate transporter, along with an increase in citrate abundance with respect to other macrophage subtypes, suggesting an important role for this pathway in TEM function. Arg1 was strongly correlated with Pik3cd, a catalytic subunit of PI3K, suggesting a role for this pathway in TEM polarization. Indeed, we also observe upregulation of several PI3K-related genes.

Next, we queried our in-house scRNA-seq dataset of human tumors, from which we observed expression of several important TEM markers in human TAMs. We also note expression of PI3K-related genes in human TAMs, indicating persistence of this pathway in macrophage polarization in clinically relevant models. As confirmation of the general understanding of TAMs, we see that proinflammatory markers are not substantially expressed in human TAMs, while anti-inflammatory markers are more abundant.

We then directed our attention to the features of pancreatic cancer cells that drive TEM polarization. Using our isogenic, dox-inducible mutant Kras PDA model, we polarized TEMs with conditioned media from either Kras-expressing or -extinguished cells. Indeed, we observed that the most

distinct markers of TEM metabolism and cytokine signaling are reliant on Kras expression in PDA cells. Kras is known to impact cancer cell glucose metabolism and growth factor expression. Specifically, lactate and GM-CSF are known to be released from cancer cells in greater abundance when mutant Kras is expressed. By querying our bulk RNA-seq dataset, we observed several GM-CSF- or lactate-responsive genes differentially expressed in Kras-On TEMs compared to Kras-Off TEMs. We then investigated how these factors may affect the expression of significant TEM markers and found that naive BMDMs treated with GM-CSF displayed increased expression of *Ccr1*, *Ccr5*, and *Acly*. Further, BMDMs treated with both lactate and GM-CSF displayed increased expression of Arg1 and Txnip. These data suggest that TEM and TAM polarization occurs in response to both metabolic crosstalk and growth factor signaling, and build upon previous reports of the role of tumor-derived lactate in TAM polarization (*Colegio et al., 2014*).

In consideration of the GM-CSF–PI3K pathway, and the strong correlation between Arg1 and Pik3cd, we treated BMDMs with either a PI3K inhibitor, AKT inhibitor, or GM-CSF-neutralizing antibody, and observed that both PI3K–AKT inhibition and GM-CSF neutralization reduced Arg1 expression relative to vehicle and IgG control, respectively. We also note changes in TEM metabolism in response to GM-CSF neutralization, most notably an increase in citrate, which may potentially be correlated with reduced *Acly* expression. Collectively, these data demonstrate an important role for mutant Kras in TEM and TAM polarization, and suggest that mutant Kras exhibits its most significant effects through increased release of GM-CSF and lactate from pancreatic cancer cells. This improved an understanding of epithelial–myeloid communication and distinct features of tumor-associated macrophages will hopefully provide new insights into potential pathways for exploitation to improve pancreatic cancer therapy.

Finally, it is important to note a few limitations in our current investigation of TEM activation in the pancreatic TME. First, the epithelial–myeloid axis provides only one node of the complex network of interactions in pancreatic tumors. In particular, fibroblasts make up a significant part of the overall cellularity of pancreatic tumors, and the heterogeneity of fibroblast populations are only now beginning to be understood (*Helms et al., 2020*; *Garcia et al., 2020*). Among these populations, many are noted to be strongly immunosuppressive, potentially providing another source of GM-CSF for the myeloid cells. Further, the pancreatic TME is characterized by poor vasculature (*Kamphorst et al., 2015*). Future studies will be needed to address the impact of the resulting hypoxia and low-nutrient availability on the epithelial–myeloid signaling axis, features that were not well recapitulated using the cell culture-based approaches presented herein.

# Materials and methods

## Cell culture

The dox-inducible (iKras*3) primary mouse PDA cell line used in this study was described previously (*Zhang et al., 2017*). Cells were maintained in high-glucose Dulbecco's modified Eagle medium (DMEM) (Gibco) supplemented with 10% fetal bovine serum (FBS) (Corning) at 37°C. iKras*3 cells were also maintained in 1 µg/ml dox. In certain conditions, iKras*3 cells were deprived of dox, for either 3 or 5 days before conditioning media, to turn mutant Kras expression off and assess Kras-dependent effects on macrophage polarization. Cells were routinely checked for mycoplasma contamination with MycoAlert PLUS (Lonza).

## Conditioned medium generation

PDA cell-conditioned medium was generated by changing the media of >50% confluent iKras*3 plates, removing media after 48 hr, and filtering through a 0.45-µm polyethersulfone membrane (VWR). Fresh media was added at a ratio of one to three parts conditioned medium to replenish nutrients consumed by cancer cells. L929-conditioned media was prepared for BMDM differentiation, as described (*Halbrook et al., 2019*). L929 mouse fibroblasts were maintained in fresh DMEM for 48 hr, after which the conditioned media was filtered through a 0.45-µm polyethersulfone membrane.

## BMDM Differentiation

Bone marrow was isolated from the femurs of C57B6/J mice as described (*Celada et al., 1984*) and maintained in macrophage differentiation media (high-glucose DMEM with 10% FBS, penicillin/

streptomycin [Gibco], sodium pyruvate [Gibco], and 30% L929-conditioned media) for 5 days. Media was refreshed on day 3, and naive macrophages were polarized on day 5.

## Macrophage polarization

BMDMs were polarized with either 10 ng/ml murine M-CSF (Peprotech), 10 ng/ml LPS (Enzo), 10 ng/ml murine IL4 (Peprotech), 2 ng/ml murine GM-CSF, or 75% Kras-On or Kras-Off PDA cell-conditioned media. In certain conditions, macrophages were spiked with 5 mM lactic acid to assess the effects of extracellular lactate and acidic pH on macrophage gene expression. Each macrophage subtype was polarized from matched biological replicates. Macrophages were maintained in the presence of polarization factors for 48 hr.

## GM-CSF neutralization and PI3K/AKT inhibition

BMDMs were differentiated over 5 days then treated for 48 hr with either 10 ng/ml murine M-CSF or 75% Kras-On PDA-conditioned media with either vehicle control, 1 nM MK-2206 (Selleck Chemicals), 1 nM BKM120 (Selleck Chemicals), 1 µg/ml anti-GM-CSF-neutralizing antibody (BioLegend), or IgG control. Compounds were maintained in dimethyl sulfoxide. Macrophages polarized in the presence of the PI3K and AKT inhibitors were pretreated with the respective compound for 30 min.

## RNA isolation and reverse transcription

Polarized BMDMs were lysed with RLT Plus buffer with β-mercaptoethanol, lysates were homogenized using a Qiashredder, and RNA samples were isolated according to the RNeasy Plus Mini Kit (Qiagen) protocol, which included gDNA eliminator spin columns. All RNA samples were tested for concentration and purity via NanoDrop (Thermo Scientific). RNA samples were stored in −80°C until needed for reverse transcription. Complementary DNA (cDNA) reverse transcription was performed following the iScript cDNA Synthesis kit protocol (BioRad), and cDNA samples were used for qPCR.

## Western blotting

Cells were lysed in radioimmunoprecipitation assay (RIPA) buffer (Sigma-Aldrich) and supplemented with phosphatase inhibitor (Sigma-Aldrich) and complete Ethylenediaminetetraacetic acid (EDTA)-free protease inhibitor (Sigma-Aldrich). Lysates were quantified by Bicinchoninic acid (BCA) assay (Thermo Fisher Scientific Inc), and equivalent protein amounts were run onto sodium dodecyl sulfate–polyacrylamide gel electrophoresis (SDS–PAGE) gels. Proteins were transferred from the SDS–PAGE gel to an Immobilon-FL PVDF membrane, blocked, and incubated with primary antibodies. After washing, membranes were incubated in secondary antibody, washed, then exposed on a Biorad Chemidoc with West Pico (Thermo Fisher Scientific) or West Femto ECL (Thermo Fisher Scientific). Quantitation was performed using Image Lab software.

| Protein | Antibody name | Catalog # | Company | Dilution |
|---------|---------------|-----------|---------|----------|
| ACLY | ATP-Citrate Lyase Antibody | #4332 | Cell Signaling | *1:1000 |
| p-ACLY | Phospho-ATP-Citrate Lyase (Ser455) Antibody | #4331 | Cell Signaling | *1:500 |
| AKT | Akt Antibody | #9272 | Cell Signaling | *1:1000 |
| p-AKT | Phospho-Akt (Ser473) (D9E) XP Rabbit mAb | #4060 | Cell Signaling | *1:1000 |
| ARG1 | Arginase-1 (D4E3M) XP Rabbit mAb | #93,668 | Cell Signaling | *1:1000 |
| TXNIP | TXNIP (D5F3E) Rabbit mAb | #14,715 | Cell Signaling | *1:1000 |
| pERK | Phospho-p44/42 MAPK (Erk1/2) (Thr202/Tyr204) (E10) Mouse mAb | #9106 | Cell Signaling | *1:1000 |
| ERK | p44/42 MAPK (Erk1/2) (137F5) Rabbit mAb | #4695 | Cell Signaling | *1:1000 |

*Continued on next page*

*Continued*

| Protein | Antibody name | Catalog # | Company | Dilution |
|---|---|---|---|---|
| Vinculin | Vinculin (E1E9V) XP | #13,901 | Cell Signaling | *1:5000 |
| Anti-rabbit IgG HRP-linked Secondary | Anti-rabbit IgG, HRP-linked Antibody | #7074 | Cell Signaling | *1:5000 |
| Anti-mouse IgG HRP-linked Secondary | Anti-mouse IgG, HRP-linked Antibody | #7076 | Cell Signaling | *1:5,000 |

## Lactate measurement

Lactate measurements were carried out using the lactate fluorescence assay kit (Biovision #K607). Assays were performed according to the manufacturer's instructions. Lactate levels were measured using a SpectraMax M3 Microplate reader (Molecular Devices).

## RNA-seq and data analysis

RNA-seq and data analysis were performed as described (*Zhang et al., 2020*). Upon isolation of RNA samples, and determination of RNA concentration and quality, the University of Michigan Sequencing Core prepared strand mRNA libraries that were sequenced using 50-cycle paired-end reads via a HiSeq 4000 (Illumina) sequencing system. Raw data were generated and analyzed by the University of Michigan Bioinformatics Core. A quality control (QC) was performed using FastQC software (Babraham Bioinformatics) for both pre- and postalignment. Raw sequencing reads were aligned to the University of California Santa Cruz (UCSC) mm10 assembly mouse genome browser with Bowtie2 and TopHat tools of the Tuxedo suite RNA-seq alignment software. Quantification of gene expression was performed with HTSeq to generate TPM values. Relative expression was graphed as mean-centered abundance, in which each sample's raw expression value was divided by the mean expression value of all samples. The primary data are available at GEO (GSE189354).

## Quantitative polymerase chain reaction

Samples for qPCR were prepared with 1× Fast SYBR Green PCR master mix (Applied Biosystems). Primers were optimized for amplification under the following conditions: 95°C for 10 min, followed by 40 cycles of 95°C for 15 s and 60°C for 1 min. Melt curve analysis was performed for all samples upon completion of amplification. Hypoxanthine phosphoribosyltransferase (*Hprt1*) primer was used as a reference gene. Relative quantification was calculated using the $2^{-\Delta\Delta CT}$ method, in which the cycle threshold (CT) value of a target sample's target gene is normalized to the expression of a reference gene in both a reference sample and the target sample.

| Gene | 5' Primer | 3' Primer |
|---|---|---|
| Acly | GAGGGGAAGCTGATCATGGG | GAGCCACAGTTCCTGAGCAT |
| Arg1 | CAGAAGAATGGAAGAGTCAG | CAGATATGCAGGGGAGTCACC |
| Ccr1 | AGGAATTGGCCACTGGTGAG | TTGCTGAGGAACTGGTCAGG |
| Ccr5 | AGACATCCGTTCCCCCTACA | GCAGCATAGTGAGCCCAGAA |
| Chil3 | CAGGGTAATGAGTGGGTTGG | CACGGCACCTCCTAAATTGT |
| Fizz1 | CCTGCTGGGATGACTGCTAC | GTCAACGAGTAAGCACAGGC |
| Gmcsf | ATGCCTGTCACGTTGAATGAAG | GCGGGTCTGCACACATGTTA |
| Gmcsf | AGATATTCGAGCAGGGTCTAC | GGGATATCAGTCAGAAAGGTT |
| Hprt1 | TCAGTCAACGGGGGACATAAA | GGGGCTGTACTGCTTAACCAG |
| Il12b | TGGTTTGCCATCGTTTTGCTG | ACAGGTGAGGTTCACTGTTTCT |
| Il1b | CGCAGCAGCACATCAACAAG | GTGCTCATGTCCTCATCCTG |

*Continued on next page*

*Continued*

| Gene | 5' Primer | 3' Primer |
|------|-----------|-----------|
| Il4i1 | GCCATTCCCCAGAGGACATC | GGCTGTACCGGAGTCTATCG |
| NOS2 | GTTCTCAGCCCAACAATACAAGA | GTGGACGGGTCGATGTCAC |
| Slc25a1 | TGCGACTGTACTGAAGCAGG | GTAGAATGCCTTTGGCCCCT |
| Slc2a1 | GTGACGATCTGAGCTACGGG | GAGAGACCAAAGCGTGGTGA |
| Tbp | CCCCACAACTCTTCCATTCT | GCAGGAGTGATAGGGGTCAT |
| Tnfa | GACGTGGAACTGGCAGAAGAG | TTGGTGGTTTGTGAGTGTGAG |
| Txnip | CCCTGACCTAATGGCACCAG | AGTGTGTCGGGCCACAATAG |

## Proteomics

### Sample preparation

Six total samples from six macrophage subtypes were prepared in duplicate for MS-based proteomics. The supernatant of each sample's cell lysate was collected to obtain >70 µg of total protein or a protein concentration of 2 µg/µl per sample. Samples were stored at −80°C until the proteomics experiments.

### Tandem mass tag (TMT) quantification

Protein identification and TMT quantification were performed using Proteome Discoverer (v2.1, Thermo Fisher Scientific). MS2 spectra were searched against *Mus musculus* protein database (UniProt, 25,510 entries, downloaded on 10/03/2017) using the following search parameters: MS1 and MS2 tolerance were set to 10 ppm and 0.6 Da, respectively; carbamidomethylation of cysteines (57.02146 Da) and TMT labeling of lysine and N-termini of peptides (229.16293 Da) were considered static modifications; oxidation of methionine (15.9949 Da) and deamidation of asparagine and glutamine (0.98401 Da) were considered variable. Percolator PSM validator was used to filter Identified proteins and peptides to retain only those that passed ≤2% FDR threshold. Quantitation was performed using high-quality MS3 spectra (average signal-to-noise ratio of 6% and <50% isolation interference). A total of 6919 proteins were quantified and 5437 proteins were common in the two TMT experiments. The mean and median of Pearson's correlation coefficients between the abundance profiles of individual proteins in both TMT datasets were 0.68 and 0.84, respectively. There were 3631 proteins whose abundance profile correlations were greater than the mean, which we considered consistent between the two TMT experiments. For downstream analysis, the mean-centered normalized data were used.

### LC-MS3 analysis

For raw data acquisition from a total of 28 runs (14 in duplicate), an Orbitrap Fusion (Thermo Fisher) and Rapid Separation Liquid Chromatography (RSLC) Systems UltiMate 3000 nano-Ultra Performance Liquid Chromatography (UPLC) (Dionex) were used. To increase accuracy and confidence in protein abundance measurements, a multinotch-MS3 analysis method was employed for MS data analysis. Two microliters from each fraction were resolved in 2D on a nanocapillary reverse phase column (Acclaim PepMap C18, 2 µm particle size, 75 µm diameter × 50 cm length, Thermo Fisher) using a 0.1% formic/acetonitrile gradient at 300 nl/min (2%–22% acetonitrile in 150 min, 22%–32% acetonitrile in 40 min, 20-min wash at 90% acetonitrile, followed by 50-min reequilibration) and sprayed directly onto the Orbitrap Fusion with EasySpray (Thermo Fisher; Spray voltage (positive ion) = 1900 V, Spray voltage (negative ion) = 600 V, method duration = 180 min, ion source type = nanoelectrospray ionization (NSI)). The mass spectrometer was set to collect the MS1 scan (Orbitrap; 120 K resolution; automatic gain control [AGC] target $2 \times 105$; max injection time [IT] 100 ms), and then data-dependent Top Speed (3 s) MS2 scans (collision-induced dissociation; ion trap; NCD 35; AGC $5 \times 103$; max IT 100 ms). For multinotch-MS3 analysis, the top 10 precursor ions from each MS2 scan were fragmented by high-energy collisional dissociation, followed by Orbitrap analysis (NCE 55; 60 K resolution; AGC $5 \times 104$; max IT 120 ms; 100–500 *m/z* scan range).

## Tandem mass tag (TMT) data analysis

Raw MS data preprocessing and TMT protein quantification were performed using MSFragger (*Kong et al., 2017*) (peptide identification), the Philosopher toolkit (*da Veiga Leprevost et al., 2020*) (peptide validation and protein inference, FDR filtering, and extraction of quantitative information from MS scans), and TMT-Integrator (protein quantification and normalization) as previously described (*Djomehri et al., 2020*). A total of 6919 proteins were quantified and 5437 proteins were common in the two TMT experiments. The mean and median of Pearson's correlation coefficients between the abundance profiles of individual proteins in both TMT datasets were 0.68 and 0.84, respectively. There were 3631 proteins whose abundance profile correlations were greater than the mean, which we considered consistent between the two TMT experiments. For downstream analysis, the mean-centered normalized data were used. The candidate markers of differentially abundant proteins for each macrophage subtype were identified by a one-tailed *t*-test for each direction of up- and/or downregulation against the remaining subtypes with a p value threshold of 0.001. No multiple testing correction was made in favor of downstream functional analysis.

The MS proteomics data have been deposited to the ProteomeXchange Consortium via the PRIDE partner repository with the dataset identifier PXD028632.

## Metabolite sample preparation

Intracellular metabolite fractions were prepared from cells grown in nontissue culture-treated 6-well plates (Corning) that were lysed with cold (−80°C) 80% methanol, then clarified by centrifugation. Metabolite levels of intercellular fractions were normalized to the protein content of a parallel sample, and all samples were dried via speed vac after clarification by centrifugation. Media samples were prepared by collecting 200 µl of conditioned or basal media and adding to 800 µl of cold 100% methanol. The resultant was clarified by centrifugation and lyophilized via speed vac. Dried metabolite pellets from cells or media were resuspended in 35 µl 50:50 HPLC grade methanol:water mixture for metabolomics analysis.

## Metabolomics

Agilent 1290 UHPLC and 6490 Triple Quadrupole (QqQ) Mass Spectrometer (LC–MS) were used for label-free targeted metabolomics analysis, as described previously (*Lee et al., 2019*). Agilent MassHunter Optimizer and Workstation Software LC–MS Data Acquisition for 6400 Series Triple Quadrupole B.08.00 was used for standard optimization and data acquisition. Agilent MassHunter Workstation Software Quantitative Analysis Version B.0700 for QqQ was used for initial raw data extraction and analysis. For RPLC, a Waters Acquity UPLC BEH TSS C18 column (2.1 × 100 mm, 1.7 µm) was used in the positive ionization mode. For HILIC, a Waters Acquity UPLC BEH amide column (2.1 × 100 mm, 1.7 µm) was used in the negative ionization mode. Further details are found in our previous study (*Lee et al., 2019*). The unprocessed metabolomics data are presented in *Supplementary file 5*.

## Bioinformatics and statistical analysis

Bioinformatics analyses were performed using R/Bioconductor. Differential expression or abundance analysis for either up- or downregulation was done using a one-tailed *t*-test for each subtype against all the others. Differential markers were identified using a p value threshold of 0.001. p values were not adjusted for multiple testing in favor of flexibility in downstream analyses and biological interpretations. Heat maps were made using R and Morpheus (https://software.broadinstitute.org/morpheus). Metabolomics pathway analyses were performed using MetaboAnalyst 5.0 (*Pang et al., 2021*). Bar plots were created using GraphPad Prism 9. GSEA was performed using GSEA 4.1.0 (*Subramanian et al., 2005*; *Mootha et al., 2003*), relevant parameters including the c2.cp.kegg.v7.4.symbols gene set, and gene names converted with Mouse_ENSEMBL_Gene_ID_Human_Orthologs_MSigDB. v7.4. Statistical analyses were performed using GraphPad Prism 9. Comparisons of two groups were analyzed using unpaired, two-tailed Student's *t*-test. Comparisons with more than two groups were analyzed with one-way analysis of variance with Tukey's post hoc test. All error bars represent mean with standard deviation.

.

## Acknowledgements

NGS was supported by K99CA26315401. YZ was supported by R50CA232985. AIN was supported by the NCI U24CA210967. MPdM and CAL were supported by U01CA224145 and UMCCC Core Grant P30CA046592. CJH was supported by F32CA228328, K99/R00CA241357, P30DK034933, UCI Cancer Center Support Grant P30CA062203, a National Pancreas Foundation Research Grant, V Scholar Award, and a Sky Foundation Research Grant. CAL was supported by the NCI (R37CA237421, R01CA248160, R01CA244931). Metabolomics studies performed at the University of Michigan were supported by NIH grant DK097153, the Charles Woodson Research Fund, and the UM Pediatric Brain Tumor Initiative.

## Additional information

### Competing interests

Costas A Lyssiotis: CAL has received consulting fees from Astellas Pharmaceuticals and Odyssey Therapeutics and is an inventor on patents pertaining to Kras regulated metabolic pathways, redox control pathways in pancreatic cancer, and targeting the GOT1-pathway as a therapeutic approach (US Patent No: 2015126580-A1, 05/07/2015; US Patent No: 20190136238, 05/09/2019; International Patent No: WO2013177426-A2, 04/23/2015). The other authors declare that no competing interests exist.

### Funding

| Funder | Grant reference number | Author |
|---|---|---|
| National Cancer Institute | R50CA232985 | Yaqing Zhang |
| National Cancer Institute | U24CA210967 | Alexey I Nesvizhskii |
| National Cancer Institute | U01CA224145 | Marina Pasca di Magliano Costas A Lyssiotis |
| National Cancer Institute | P30CA046592 | Marina Pasca di Magliano Costas A Lyssiotis |
| National Cancer Institute | F32CA228328 | Christopher J Halbrook |
| National Cancer Institute | K99/R00CA241357 | Christopher J Halbrook |
| National Institute of Diabetes and Digestive and Kidney Diseases | P30DK034933 | Christopher J Halbrook |
| National Cancer Institute | P30CA062203 | Christopher J Halbrook |
| National Pancreas Foundation | NPF-5615796 | Christopher J Halbrook |
| V Foundation for Cancer Research | V2021-026 | Christopher J Halbrook |
| Sky Foundation | | Christopher J Halbrook |
| National Cancer Institute | R37CA237421 | Costas A Lyssiotis |
| National Cancer Institute | R01CA248160 | Costas A Lyssiotis |
| National Cancer Institute | R01CA244931 | Costas A Lyssiotis |
| National Institute of Diabetes and Digestive and Kidney Diseases | DK097153 | Costas A Lyssiotis |
| Charles Woodson Research Fund | | Costas A Lyssiotis |
| UM Pediatric Brain Tumor Initiative | | Costas A Lyssiotis |
| National Cancer Institute | K99CA26315401 | Nina Steele |

| Funder | Grant reference number | Author |
|---|---|---|

The funders had no role in study design, data collection, and interpretation, or the decision to submit the work for publication.

## Author contributions

Seth Boyer, Conceptualization, Data curation, Formal analysis, Investigation, Writing – original draft, Writing – review and editing; Ho-Joon Lee, Conceptualization, Data curation, Formal analysis, Investigation, Methodology, Project administration, Software, Supervision, Writing – review and editing; Nina Steele, Data curation, Formal analysis, Investigation, Writing – review and editing; Li Zhang, Matthew H Ward, Formal analysis, Investigation, Writing – review and editing; Peter Sajjakulnukit, Rima Singh, Formal analysis, Investigation; Anthony Andren, Data curation, Investigation; Venkatesha Basrur, Data curation, Formal analysis, Resources, Writing – review and editing; Yaqing Zhang, Data curation, Formal analysis, Writing – review and editing; Alexey I Nesvizhskii, Data curation, Methodology, Resources, Supervision, Writing – review and editing; Marina Pasca di Magliano, Funding acquisition, Investigation, Writing – review and editing; Christopher J Halbrook, Conceptualization, Data curation, Formal analysis, Investigation, Methodology, Project administration, Supervision, Writing – review and editing; Costas A Lyssiotis, Conceptualization, Funding acquisition, Project administration, Resources, Supervision, Writing – original draft, Writing – review and editing

## Author ORCIDs

Ho-Joon Lee http://orcid.org/0000-0003-3616-5387
Matthew H Ward http://orcid.org/0000-0001-7135-5878
Rima Singh http://orcid.org/0000-0002-6822-7987
Christopher J Halbrook http://orcid.org/0000-0002-3376-3114
Costas A Lyssiotis http://orcid.org/0000-0001-9309-6141

## Decision letter and Author response

Decision letter https://doi.org/10.7554/eLife.73796.sa1
Author response https://doi.org/10.7554/eLife.73796.sa2

# Additional files

## Supplementary files

• Supplementary file 1. Differential expression or abundance analysis detailing the markers that distinguish each subtype from *Figure 1E*.

• Supplementary file 2. Gene set enrichment analysis (GSEA) of KEGG gene sets from the RNA-seq dataset comparing tumor-educated macrophages (TEMs) to M0, M1, and M2 macrophages.

• Supplementary file 3. Enrichr analysis of the proteomics dataset comparing tumor-educated macrophages (TEMs) to M0, M1, and M2 macrophages.

• Supplementary file 4. Gene set enrichment analysis (GSEA) of KEGG gene sets from tumor-educated macrophages (TEMs between the Kras-On and Kras-Off) conditions.

• Supplementary file 5. Raw metabolomics data from *Figure 1* and *Figure 5—figure supplement 1D*.

• Transparent reporting form

• Source data 1. Raw western blot images.

## Data availability

RNA-seq data were deposited to NIH GEO (https://www.ncbi.nlm.nih.gov/geo/) with the dataset identifier GSE189354; the mass spectrometry proteomics data have been deposited to the ProteomeXchange Consortium via the PRIDE (https://www.ebi.ac.uk/pride/) partner repository with the dataset identifier PXD028632. Metabolomics data from Figure 1 and Figure 5-Figure Supplement 1A are included in Supplementary file 5. Annotated raw blots are included for all Westerns as Source Data.

The following datasets were generated:

| Author(s) | Year | Dataset title | Dataset URL | Database and Identifier |
|-----------|------|---------------|-------------|-------------------------|
| Lyssiotis CA | 2022 | Multi-omic Characterization of Pancreatic Cancer-Associated Macrophage Polarization Reveals Deregulated Metabolic Programs Driven by the GMCSF-PI3K Pathway | http://www.ncbi.nlm.nih.gov/geo/query/acc.cgi?acc=GSE189354 | NCBI Gene Expression Omnibus, GSE189354 |
| Lyssiotis CA | 2022 | Multi-omic Characterization of Pancreatic Cancer-Associated Macrophage Polarization Reveals Deregulated Metabolic Programs Driven by the GMCSF-PI3K Pathway | https://www.ebi.ac.uk/pride/archive/projects/PXD028632 | PRIDE, PXD028632 |

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
