## [Editor Report]

This paper performs a comprehensive mechanistic and genomic evaluation of the impact of macrophage polarization on metabolic changes in pancreatic cancer. It provides an important advance to the understanding of the role of the microenvironment in the context of this disease.

---

## [Decision Letter]

**Decision letter after peer review:**

Thank you for submitting your work entitled "Multi-omic Characterization of Pancreatic Cancer-Associated Macrophage Polarization Reveals Deregulated Metabolic Programs Driven by the GMCSF-PI3K Pathway" for consideration by *eLife*. Your article has been reviewed by 3 peer reviewers, one of whom is a member of our Board of Reviewing Editors, and the evaluated has been overseen by Mone Zaidi as the Senior Editor. The following individual involved in review of your submission has agreed to reveal their identity: David DeNardo (Reviewer #3).

The manuscript has been improved but there are some remaining issues that need to be addressed, as outlined below:

*Reviewer #1:*

This paper performs comprehensive multi-omics assessment of tumor-educated macrophages to evaluate the mechanisms underlying their critical role in pancreatic cancer.

While this is unquestionably an important study for the field and these omics data are comprehensive, the bioinformatics analysis and data presentation could benefit from further enhancement to better convey the biological findings from the data.

The primary critique in this paper is the presentation and analyses based upon the omics data. These need to be greatly enhanced for publication as described below.

Figure 1 – Differential analyses refer to performing a one-way anova with dunnett's post-hoc test, but the methods describe DESeq2 analysis for the RNA-seq data. This should be clarified in the figure. The presentation for panels C / D would benefit from lines over the *** to clarify that comparisons are relative to the TEM population. In the heatmaps in D, it's unclear why the particular group of TEM is boxed or what comparisons are made to obtain the subset of genes/proteins that are plotted. Moreover, in absence of gene names these heatmaps do not provide considerable information. Recommend adding volcano plots, considering annotations of genes, or similar to help obtain biological insights from the heat maps presented. At minimum, a supplemental table of the differential expression analysis would enable interested readers to evaluate the findings. IT is also unclear whether the mean centering for the heatmaps is performed on variance stabilized gene expression data or merely log transformed data.

The subsection "Metabolism and cytokine signaling are distinctive features of pancreatic TEMs" describes pathway-centric approaches but surprisingly does not perform any pathway analyses.

Figure 2 – As described for figure 1, the statistical method described in the figure caption for the RNA-seq data does not match the methods and would benefit from improved visualization to clarify that comparisons are made relative to to the TEM group. Red / green coloring should be avoided in the heatmap in panel E.

The STRING analysis refers to "a particularly strong functional association", but does not provide statistics to describe how strength was assessed.

Individual PI3K pathway genes are assessed in the scRNA-seq data, but pathway analyses would strengthen the proposed associations.

Figure 4 – As described above, the heatmap could benefit from enhanced gene annotation to assess the biological relevance of the analyses performed.

*Reviewer #2:*

The authors find that tumor cell conditioned media from Kras-expressing PDAC cells contains high levels of GM-CSF and lactate, which combine to generate macrophages that resemble suppressive macrophages found in the PDAC tumor microenvironment. The authors perform extensive metabolomics on cultured macrophages, correlate these findings with transcriptional and protein-level data, and connect the dots between Kras, GM-CSF, PI3K/AKT signaling, and the suppressive enzyme Arg1. They also find that GM-CSF and lactate are sufficient to synergistically induce Arg1 transcription.

Overall, these data support and extend a wealth of prior literature showing suppressive effects of GM-CSF on tumor-associated myeloid cells, including two seminal papers in Cancer Cell in 2012 reporting that pancreatic cancer cells secreted high levels of GM-CSF. GM-CSF can come from a number of sources in the tumor microenvironment, notably fibroblasts (PMID: 27184426), which are not modeled in the work as currently presented and should be mentioned in the discussion as a limitation/opportunity for future directions.

There are several limitations to the work that modestly detract from the authors' conclusions. First, the in vitro studies were performed in standard tissue culture conditions and the contribution of hypoxia was not addressed. This should be mentioned in the discussion in a paragraph about ways to improve the cross-talk in vitro model along with inclusion of other cell types such as fibroblasts. Second, the connection to human PDAC macrophages is tenuous. Examination of Supplemental Figure 3 reveals that both TXNIP and ACLY are robustly expressed across most cell types, and indeed all of the TEM macrophage genes shown here are shared with at least one other cell type. I believe the authors are trying to say that human PDAC macrophages show a signature of having been polarized by GM-CSF and lactate. It is possible that the other cell types in the tumor microenvironment are not the best comparison; a better strategy might be to compare macrophages differentiated from peripheral blood CD14+ cells +/- human PDAC conditioned media. Alternatively, the authors could examine macrophages from their chronic pancreatitis cohort (PMID: 34296197) or other non-PDAC tissue macrophages as a comparison to show the PDAC-induced upregulation of a TEM signature.

Line 140 and Figure 2B: What is an "enzyme marker"? Please define the specific enzymes that are differentially regulated in TEMs.

Lines 261-263: "In addition, blocking GM-CSF led to a modest decrease in expression of ACLY and TXNIP (Figure 6B, Supplementary Figure 5B), which builds upon our M0 + GM-CSF + lactate qPCR data, in support of an activating role of GM-CSF on enzymatic TEM markers." Awkward phrasing. Please rewrite for clarity and remove the phrase "enzymatic TEM markers".

Figure 6 legend: Please include a reminder to the reader what the drugs target (pan-PI3K inhibitor, BKM120; pan-AKT inhibitor, MK-2206).

Lines 266-267: GM-CSF blockade appears to have no effect on phosphorylated sugars as shown in Supplemental Figure 5C. Please modify this statement accordingly.

*Reviewer #3:*

The paper by Boyer et al., is an excellent multi-omic approach to try to link changes in macrophage metabolism with RNAseq and proteomic changes. The authors show PDAC-cell derived cytokines (GM-CSF) along with metabolic mediators (lactate) regulate key features in macrophage phenotype and metabolism. This is overall an excellent study with high impact. I have no formal suggestion of new experiments needed, as the studies already done are excellent. However the paper would be considerable improved for the reader with some changes in text and some detailed analysis of the data sets on hand. Otherwise the authors are to commended on a fine piece of work.

1. The authors have a very nice data set. I would love to see them run traditional pathway analysis, GSEA on proteomics and RNAseq, and layer this into the story line. There are some hints at this with limited string analysis and Figure 2C. But this could be improved and discussed. These are likely biologic process that TEMs have that require the metabolic switched shown. Anything here would be of interest.

2. In Figure 4, agin the data are excellent. But I would really love to see more formal pathway analysis and GSEA approaches to identify biologic process. And then I would love to see the authors comment on how these changes in biologic process, require or cross talk with the metabolism. Of course, this can take place in analysis added to results and commentary in the discussion.

3. There is some ambiguity in the results presented in Figure 2B-ED The legend and figure say genes and proteins. Which is it. Did the authors look for overlapping changes (ven diagram style) and then map these to the results? The authors should be explicit here what they did and it they are showing RNA or protein data. E appears to say co-expressed RNA/protein, but details lacking.

---

## [Author Response]

Reviewer #1:This paper performs comprehensive multi-omics assessment of tumor-educated macrophages to evaluate the mechanisms underlying their critical role in pancreatic cancer.While this is unquestionably an important study for the field and these omics data are comprehensive, the bioinformatics analysis and data presentation could benefit from further enhancement to better convey the biological findings from the data.

We thank the reviewer for their comments and support of our work. In the point-by-point response that follows, we have addressed the excellent suggestions in full, which we believe have provided additional clarity and improved the manuscript.

The primary critique in this paper is the presentation and analyses based upon the omics data. These need to be greatly enhanced for publication as described below.Figure 1 – Differential analyses refer to performing a one-way anova with dunnett's post-hoc test, but the methods describe DESeq2 analysis for the RNA-seq data. This should be clarified in the figure.

We thank the referee for catching this oversight and appreciate the opportunity to make the correction. We did not use DESeq2 results in this work; the methods have been updated for accuracy. The one-way ANOVA with Dunnett’s post-hoc test was done only for individual box plots, as we have now clarified in the legend of Figure 1.

The presentation for panels C / D would benefit from lines over the *** to clarify that comparisons are relative to the TEM population.

We appreciate this useful suggestion. Significance is presented relative to the TEM subtype. This has been clarified in the updated legend, as well as in the figure using lines to directly indicate comparisons, as suggested.

In the heatmaps in D, it's unclear why the particular group of TEM is boxed or what comparisons are made to obtain the subset of genes/proteins that are plotted. Moreover, in absence of gene names these heatmaps do not provide considerable information. Recommend adding volcano plots, considering annotations of genes, or similar to help obtain biological insights from the heat maps presented. At minimum, a supplemental table of the differential expression analysis would enable interested readers to evaluate the findings.

We appreciate the opportunity to clarify this point. The boxed groups in the heatmaps represent “TEM markers” for each omics dataset, so as to highlight the reader’s focus on TEM signatures in this work. This is now stated in the legend. In addition, as suggested, we have now provided all markers in the heatmaps as Supplementary File 1.

IT is also unclear whether the mean centering for the heatmaps is performed on variance stabilized gene expression data or merely log transformed data.

We again thank the referee for bringing this matter for clarification to our attention. Mean centering was performed for TPM values, which has now also been updated in the Methods.

The subsection "Metabolism and cytokine signaling are distinctive features of pancreatic TEMs" describes pathway-centric approaches but surprisingly does not perform any pathway analyses.

We have now performed pathway analyses by GSEA and Enrichr, as mentioned at Line 124-134. Full results are provided as Figure 1—figure supplement 1D and Supplementary Files 2,3.

Figure 2 – As described for figure 1, the statistical method described in the figure caption for the RNA-seq data does not match the methods and would benefit from improved visualization to clarify that comparisons are made relative to to the TEM group. Red / green coloring should be avoided in the heatmap in panel E.

Please see our response above regarding the statistical method. We have now changed the heatmap coloring to red/blue in Figure 2E.

The STRING analysis refers to "a particularly strong functional association", but does not provide statistics to describe how strength was assessed.

We have now added the enrichment p-value of ~0.0001 from the STRING analysis (see line 169-171).

Individual PI3K pathway genes are assessed in the scRNA-seq data, but pathway analyses would strengthen the proposed associations.

We have now performed a pathway analysis of those PI3K-related genes using Enrichr, as stated at Line 183-184 and presented in Supplementary File 3.

Figure 4 – As described above, the heatmap could benefit from enhanced gene annotation to assess the biological relevance of the analyses performed.

We agree with this point, and noted in our response above, we now provide a full list of genes in the heatmap in Supplementary File 1.

Reviewer #2:The authors find that tumor cell conditioned media from Kras-expressing PDAC cells contains high levels of GM-CSF and lactate, which combine to generate macrophages that resemble suppressive macrophages found in the PDAC tumor microenvironment. The authors perform extensive metabolomics on cultured macrophages, correlate these findings with transcriptional and protein-level data, and connect the dots between Kras, GM-CSF, PI3K/AKT signaling, and the suppressive enzyme Arg1. They also find that GM-CSF and lactate are sufficient to synergistically induce Arg1 transcription.Overall, these data support and extend a wealth of prior literature showing suppressive effects of GM-CSF on tumor-associated myeloid cells, including two seminal papers in Cancer Cell in 2012 reporting that pancreatic cancer cells secreted high levels of GM-CSF. GM-CSF can come from a number of sources in the tumor microenvironment, notably fibroblasts (PMID: 27184426), which are not modeled in the work as currently presented and should be mentioned in the discussion as a limitation/opportunity for future directions.

We appreciate the reviewer’s enthusiasm for our work. We also agree that, while we have found that GM-CSF can be released from cancer cells, this neither demonstrates that the cancer cells are the dominant source in the tumor microenvironment, nor does it preclude other sources. We have added this to our discussion and agree that this will be an important direction for future study (see lines 338-343).

There are several limitations to the work that modestly detract from the authors' conclusions. First, the in vitro studies were performed in standard tissue culture conditions and the contribution of hypoxia was not addressed. This should be mentioned in the discussion in a paragraph about ways to improve the cross-talk in vitro model along with inclusion of other cell types such as fibroblasts.

This is an excellent point. We agree with the reviewer that cell culture imposes several non-physiological restraints on the systems used in this study. As suggested, we have updated the discussion accordingly (lines 343-347). We detail how these limitations may impact the interpretation of our findings, as well as how our work can now serve as a basis for future studies that assess the impact of these variables.

Second, the connection to human PDAC macrophages is tenuous. Examination of Supplemental Figure 3 reveals that both TXNIP and ACLY are robustly expressed across most cell types, and indeed all of the TEM macrophage genes shown here are shared with at least one other cell type. I believe the authors are trying to say that human PDAC macrophages show a signature of having been polarized by GM-CSF and lactate. It is possible that the other cell types in the tumor microenvironment are not the best comparison; a better strategy might be to compare macrophages differentiated from peripheral blood CD14+ cells +/- human PDAC conditioned media. Alternatively, the authors could examine macrophages from their chronic pancreatitis cohort (PMID: 34296197) or other non-PDAC tissue macrophages as a comparison to show the PDAC-induced upregulation of a TEM signature.

We thank the reviewer for this insightful point and excellent suggestion. We leveraged our “normal” (i.e. macrophages in nearby/adjacent normal tissue collected during pancreatic tumor resections; Steele, et al., *Nature Cancer* 2020) to determine the expression profile of our TEM signature. This is now presented in Figure 3D. In brief, we observed that the expression of 9 out of 11 markers identified in our murine system are also increased in human TAMs, relative to the adjacent normal macrophages.

Line 140 and Figure 2B: What is an "enzyme marker"? Please define the specific enzymes that are differentially regulated in TEMs.

We recognize how this terminology may have caused confusion and thank the referee for bringing it to our attention. We have updated the manuscript text to indicate that our analyses pursued TEM markers that are metabolic enzymes. These are listed in the table in Figure 2C.

Lines 261-263: "In addition, blocking GM-CSF led to a modest decrease in expression of ACLY and TXNIP (Figure 6B, Supplementary Figure 5B), which builds upon our M0 + GM-CSF + lactate qPCR data, in support of an activating role of GM-CSF on enzymatic TEM markers." Awkward phrasing. Please rewrite for clarity and remove the phrase "enzymatic TEM markers".

As per the reviewer's suggestion, we have re-written this section to improve the clarity of our findings.

Figure 6 legend: Please include a reminder to the reader what the drugs target (pan-PI3K inhibitor, BKM120; pan-AKT inhibitor, MK-2206).

We appreciate this suggestion and have added this information to the legend of Figure 6.

Lines 266-267: GM-CSF blockade appears to have no effect on phosphorylated sugars as shown in Supplemental Figure 5C. Please modify this statement accordingly.

We agree that these data were not informative and have removed them from the revised manuscript.

Reviewer #3:The paper by Boyer et al., is an excellent multi-omic approach to try to link changes in macrophage metabolism with RNAseq and proteomic changes. The authors show PDAC-cell derived cytokines (GM-CSF) along with metabolic mediators (lactate) regulate key features in macrophage phenotype and metabolism. This is overall an excellent study with high impact. I have no formal suggestion of new experiments needed, as the studies already done are excellent. However the paper would be considerable improved for the reader with some changes in text and some detailed analysis of the data sets on hand. Otherwise the authors are to commended on a fine piece of work.

We are grateful for the reviewer’s compliments and enthusiasm for this study. Their suggestions have been incorporated into this revised manuscript, and we believe that these have significantly improved the accessibility of the study.

1. The authors have a very nice data set. I would love to see them run traditional pathway analysis, GSEA on proteomics and RNAseq, and layer this into the story line. There are some hints at this with limited string analysis and Figure 2C. But this could be improved and discussed. These are likely biologic process that TEMs have that require the metabolic switched shown. Anything here would be of interest.

We agree and appreciate this helpful suggestion. We have now performed KEGG pathway analyses by GSEA for the transcriptomic data and Enrichr for the proteomic data, with the results detailed starting at Line 125. This suggestion proved especially helpful to bridge our previous observations and the new results in this manuscript. Full results of these pathway analyses are provided in Figure 1—figure supplement 1D and as Supplementary Files 2,3.

2. In Figure 4, agin the data are excellent. But I would really love to see more formal pathway analysis and GSEA approaches to identify biologic process. And then I would love to see the authors comment on how these changes in biologic process, require or cross talk with the metabolism. Of course, this can take place in analysis added to results and commentary in the discussion.

We again agree and appreciate the suggestion. The formal pathway analyses contrasting the full set of macrophage subtypes, including those in Figure 4, are provided in Supplementary files 1,3. A GSEA analysis specific to the KrasOn vs. KrasOff macrophage subtypes in Figure 4 is now discussed in the text with select enrichment plots presented in Figure 4—figure supplement 1A. The full results are presented in Supplementary file 4.

3. There is some ambiguity in the results presented in Figure 2B-ED The legend and figure say genes and proteins. Which is it. Did the authors look for overlapping changes (ven diagram style) and then map these to the results? The authors should be explicit here what they did and it they are showing RNA or protein data. E appears to say co-expressed RNA/protein, but details lacking.

We are sorry for the confusion and appreciate the opportunity to clarify these results. Figure 2B contains metabolic enzymes identified as being unique for the given macrophage subtype in both the protein and gene analyses. Figure 2C contains a list of those markers (from Figure 2B) that are unique to TEMs. Figure 2D presents transcript levels from the RNA-seq, and Figure 2E is proteomic data. In the revised manuscript, we have updated the legend and associated text to clarify the biomolecule(s) presented.